# Temporary childbirth migration and maternal health care in India

**Nadia Diamond-Smith**[1]*, **Lakshmi Gopalakrishnan**[2], **Sumeet Patil**[3], **Lia Fernald**[2], **Purnima Menon**[4], **Dilys Walker**[5], **Alison M. El Ayadi**[1,5]

1 Department of Epidemiology and Biostatistics, University of California, San Francisco, California, United States of America, 2 School of Public Health, University of California, Berkeley, California, United States of America, 3 NEERMAN, Mumbai, India, 4 International Food Policy Research Center, New Delhi, India, 5 Department of Obstetrics, Gynecology, & Reproductive Sciences, University of California, San Francisco, California, United States of America

* nadia.diamond-smith@ucsf.edu

**Data Availability Statement:** The data underlying this study are publicly available at Harvard Dataverse (https://dataverse.harvard.edu/dataset.xhtml?persistentId=doi:10.7910/DVN/C2QFOA).

## Abstract

### Background

Women in South Asia often return to their natal home during pregnancy, for childbirth, and stay through the postpartum period—potentially impacting access to health care and health outcomes in this important period. However, this phenomenon is understudied (and not even named) in the demographic or health literature, nor do we know how it impacts health.

### Objective

The aim of this study is to measure the magnitude, timing, duration, risk factors and impact on care of this phenomenon, which we name Temporary Childbirth Migration.

### Methods

Using data from 9,033 pregnant and postpartum women collected in 2019 in two large states of India (Madhya Pradesh and Bihar) we achieve these aims using descriptive statistics and logistic regression models, combined with qualitative data from community health workers about this practice.

### Results

We find that about one third of women return to their natal home at some point in pregnancy or postpartum, mostly clustered close to the time of delivery. Younger, primiparous, and non-Hindu women were more likely to return to their natal home. Women reported that they went to their natal home because they believed that they would receive better care; this was born out by our analysis in Bihar, but not Madhya Pradesh, for prenatal care.

### Conclusions

Temporary childbirth migration is common, and, contrary to expectations, did not lead to disruptions in care, but rather led to more access to care.

**Funding:** This study was funded by Grant No.
OPP1158231 from Bill and Melinda Gates
Foundation (BMGF) to the University of California,
San Francisco and University of California,
Berkeley.

**Competing interests:** The authors have declared
that no competing interests exist.

## Contribution

We describe a hitherto un-named, underexplored yet common phenomenon that has implications for health care use and potentially health outcomes.

## 1. <u>Introduction</u>

A widespread, yet undocumented, practice among pregnant women, particularly in South Asia, is to return to their natal home during pregnancy, at the time of childbirth, or postpartum. However, a name for this phenomenon is absent from the literature and there is minimal characterization or impact of this traditional practice. Aside from a handful of small studies and anecdotal reports, we do not know how many women practice this, at which time points in their pregnancies they do this, or for how long they typically stay in their natal home. We do not know what types of women are more likely to do this or the range of reasons that underlie this process. Most importantly, we do not know what the impact of women's return to their natal home is on their care seeking behaviours or access to peripartum care, or, ultimately, the impact on maternal and child health outcomes.

Fitting what we know about this concept into the terminology of migration is challenging. Existing frameworks on migration describe permanent or temporary residence change for economic, political, environmental or social reasons, with residence change for visits to relatives or medical treatment purposively excluded [1]. The limitations of current definitions in this area are criticized, with calls for expansion of typologies to a level of nuance and flexibility capable of better capturing the relevant individual and population-level factors affecting migrant experiences. However, other authors operationalize migration more broadly, referring to any move away from the regular place of residence [2]. Thus this phenomenon may be aligned with temporary migration, defined as "migration for a specific motivation and purpose with the intention to return to the habitual residence after a limited period of time", for the purposes of childbirth [3]. With this in mind, for the purposes of this paper we refer to women's temporary relocation to their natal home during the peripartum period as "temporary childbirth migration."

Anecdotal evidence suggests that this practice also occurs elsewhere, including in Africa [4]. We focus on the phenomenon in India for this analysis because common socio-cultural practices in South Asia, including patrilocality, with most couples co-residing with or near the husband's family (marital family), and arranged marriages often being made into different villages, districts, and even states, make it more likely that women engaging in temporary childbirth migration will be outside of the reach of their usual health care.

### 1.1 Background

The majority of Indian women (73%) of reproductive age (15–49 years) are married [5]. In the socio-cultural context of India, marriage defines the rights and privileges that women have in society [6,7]. Marriage practices define women's roles and responsibilities, her place in the family, and her rank and privileges after marriage. In much of India, a woman's family after marriage is defined as her husband's family (marital family/home). Depending on a woman's community and region, most marital arrangements are completely exogamous (marrying outside her village without familial connections). India today remains predominantly a patrilocal society, where women move into their husband's household after marriage, co-residing with his parents, siblings and their families [8]. Many women are married into households outside

of the village, district, or even state of their birth, and thus a married woman often lives far from her natal family [9,10].

Among North Indian Hindus (the region of focus in this paper), this clan-exogamy and patrilocality cause a noticeable break in women's family environment from natal family to the marital family [7]. This is less pronounced in Muslim families where women tend to retain some of their natal relationships because of a prior relationship with the marital family [7]. However, women's social status similarly declines as they move into their marital home and enter a new life role as a wife with specific expectations about childbearing [6].

The norms behind fertility and motherhood are contingent on the social environment that women experience after their marriage. The marital family often has control over women's reproduction, which influences care seeking practices for prenatal, delivery, and postnatal services [6,11]. Women's autonomy has been found to be positively associated with use of pregnancy care services [12]. For women in rural North India, this could mean that women with higher autonomy are more able to seek more familiar natal environments during last trimester of their pregnancies. Previous research has documented the linkages between cultural and traditional practices and customs related to returning to natal families during pregnancy, childbirth and postnatal period in India [13–17].

The mention of temporary childbirth migration is almost absent from the literature; it appears only as a brief reference within a limited number of studies. A recent systematic review of childbirth beliefs and practices in Asia did not mention temporary childbirth migration [18]. We identified only one study in India by Gawde et al (2016) that quantitatively described temporary childbirth migration [19]. This mixed-methods study interviewed 234 women in Mumbai who had delivered in the last two years, all of whom were internal migrants (migrating within India). Gawde et al. measured how long during pregnancy a woman was "away" at her natal home and whether the delivery occurred in her natal home or current location, comparing recent to longer-term migrants. They found that two-thirds of migrant women returned to their natal home for delivery, with recent migrants spending longer periods of time away; half of recent migrants spent the majority of their pregnancies at their natal home. Qualitatively, reasons for return to the natal home included having more care and support, being expected to work less, and lower health-related costs. However, participants did express some concerns about poorer quality of services at the natal home. While this study is one of the only to describe this phenomenon, it is limited in its focus on rural to urban migrants in one city and has a relatively small sample size.

The remainder of the studies that mention temporary childbirth migration in South Asia are qualitative. The most substantial discussion of this phenomenon is by Raman et al. (2014) in urban Bangalore, India [20]. Interviews with women who delivered in the last two years found that women highly valued the support of their own mothers during pregnancy, childbirth and postpartum. Mothers were seen as the most important source of support, and it was perceived as sad when women could not be with their own mothers during this time. The amount of time women reported staying at their natal homes differed and variability was based on family circumstances. Husbands also visited their wives at their natal homes, and women noted that not only the support of their mothers, but also the support of their friends and extended family in the natal homes was critical. Another study in Tamil Nadu described similar trends of women returning to their natal home during the peripartum period, sometimes for "many months at a time", adding that this was because the woman's parents were expected to cover the costs of birth in addition to providing care [14]. Finally, a qualitative study in Bangladesh noted that women preferred to give birth in their natal home, but that financial barriers limited this option for women living in slums [21]. Other studies of traditional practices around pregnancy or delivery mentioned that this practice helped provide

mothers sufficient rest, a break from their routine household chores, ward-off potential infection from visitors, or that it was due to views about 'ritual pollution' at childbirth [11,22–25]. In Indian society, childbirth is a 'polluting' event in which both the mother and the child are considered to be impure or 'untouchable' for about 40 days after birth. It is based on the belief that the birth releases the nine months' of dirt ('narak'), or vaginal blood, that comes out of the mother's body at birth which is perceived to cause sickness or ill health [26,27]. One study from a tribal district in Maharashtra found that mother and baby were considered 'polluted', and should only be touched by close female relatives including grandmothers, traditional birth attendants and mothers-in-law [28]. Though there is no published statistic of this belief, the concept of pollution has been documented as a major cultural practice during childbirth in other Asian settings including Nepal, Pakistan, and Papua New Guinea [18,29,30].

With the predominantly qualitative literature suggesting that women who temporarily migrate for childbirth do so for the increased support provided at their natal home, temporary childbirth migration has the potential to significantly impact women's perinatal healthcare experiences through either improving the quality of perinatal care received or creating important discontinuities in perinatal care. Existing literature is limited in addressing this question globally. In Mumbai, Gawde et al. (2016) found that women who returned home reported similar prenatal care use as those who did not; however, women who delivered at their natal home had a 77% reduced odds of facility delivery [19]. Women who returned home; however, were more likely to be visited by a health care worker during the antenatal period. Aside from this, we know little about the impact on health, and thus, an important question about temporary childbirth migration is its impact on health care utilization and outcomes for women and infants.

## 1.2 Migration and maternal health

Female migration is common in India. Women make up 83% of India's permanent internal migrants, and 84% of those are marriage migrants [8]. Women also migrate for other reasons such as economic opportunity; however, economic migration only represents 1.1% of women's migration, consistent with the generally low female labor force participation in India (20.8%) [31–33].

Understanding the health consequences of migration is complicated, as they vary substantially across migrant type, legal status, and other factors [34]. Cross-national immigrants in other settings notably experience a temporary health advantage relative to similar national populations termed the "healthy migrant effect", despite health care access barriers, environmental and exposures, and behavioral adaptation [35–37]. However, populations represented within this research are often more stable migrant populations, excluding temporary migrants and clandestine workers, whose health status is extremely vulnerable [38]. Internal economic migrants also have heterogeneous experiences and health consequences, with socioeconomic status and social support critically important factors in their health trajectories [35,39]. Migration for females has unique individual opportunities and costs, including specific health vulnerabilities and needs, when compared to men, including around pregnancy and childbirth [40]. This is especially true because peak migration age coincides with peak reproductive age for women, and reproduction requires access to health services.

Much of the research in India suggests that female migrants, especially recent migrants, are less likely to meet key components of the perinatal continuum of care, including meeting the recommended number of antenatal care visits, delivering in a facility, and receiving postnatal care, when compared to non-migrant women [41–43]. One study looking at short-term migrants found similarly reduced use of maternal health services) [44]. However, women

migrating from rural to urban areas are generally found to have better outcomes than women "left behind" in rural areas [45]. This could be due to migrant selection, where those that migrate are already more likely to have healthier behaviors or outcomes. However, a migrant's ability to take advantage of services in new settings may also be based on their ability to assimilate. In a mixed-methods study in urban Lucknow among rural to urban migrant women, migrant women reported poorer respectful care from providers during childbirth than nonmigrants [46]. In this study, migrant women explained that lack of availability and experience with services in their rural home communities, combined with social norms around care, contributed to underutilization of these services even after they migrated.

### 1.3 Care across the perinatal period in India

India has one of the largest government-led Community Health Worker (CHW) program globally with over two million trained CHWs. Two cadres of all-female CHWs work at the community-level and deliver services to pregnant, lactating women and infants: (a) the Accredited Social Health Activist (ASHA) under the National Health Mission and (b) the Anganwadi Worker (AWW) under the national nutrition program, the Integrated Child Development Services (ICDS) [47,48]. ASHAs are honorary volunteer workers who receive a performance-based incentive by the type of services they provide [49,50]. AWWs are paid workers through a network of early childhood development and feeding centres, Anganwadi Centres [47]. ASHAs and AWWs generally work together to deliver essential nutrition and health services to beneficiaries living in their catchment areas (typically, population of 800–1,000). CHW care is intended to reach all residents within their catchment area; however, it is possible that women temporarily migrating for childbirth who are only staying for shorter time periods in the catchment area may be missed, leaving them at risk of care discontinuity.

In 2012, the government introduced a program to strengthen the ICDS policy framework and systems titled the *ICDS Systems Strengthening, and Nutrition Improvement Program (ISS-NIP)* across 162 high undernutrition burden districts in Uttar Pradesh, Madhya Pradesh, Andhra Pradesh, Bihar, Rajasthan, Chhattisgarh, Jharkhand, and Maharashtra to ensure greater focus on children under three years of age. This program consisted of four components, (i) ICDS institutional and systems strengthening; (ii) community mobilization and behavior change communication; (iii) convergent nutrition actions; and (iv) project management, monitoring and evaluation. As part of the ISSNIP initiative, the government launched a digital health intervention, the ICDS-CAS (Common Application Software) for Anganwadi Workers working under the ICDS system. This digital health intervention digitized paper registries maintained by AWWs, digitally entering beneficiary records allowing longitudinal tracking of beneficiaries across different life-stages. It also had reminders for upcoming home visits to beneficiaries' homes for counselling, auto generation of growth monitoring charts, and tracking immunizations for children and pregnant women.

In this paper, we use a mixed-methods approach to understand trends, risk factors and impacts of temporary childbirth migration in two states of India. First, we describe patterns of women's displacement across the perinatal period and their access to community health workers while in their natal home. Then we explore socio-demographic predictors of temporary childbirth migration and women's stated reasons for migrating. Finally, we estimate the impact of being in the natal home, compared to marital home, on prenatal, delivery, and postpartum care. We hypothesize that natal home migration will lead to a disruption in the continuum of care and therefore lower access and utilization of services (Fig 1). An alternate hypothesis in the opposite direction, is that women could receive better care at their natal home. We extend the existing primarily qualitative and descriptive literature by using a large

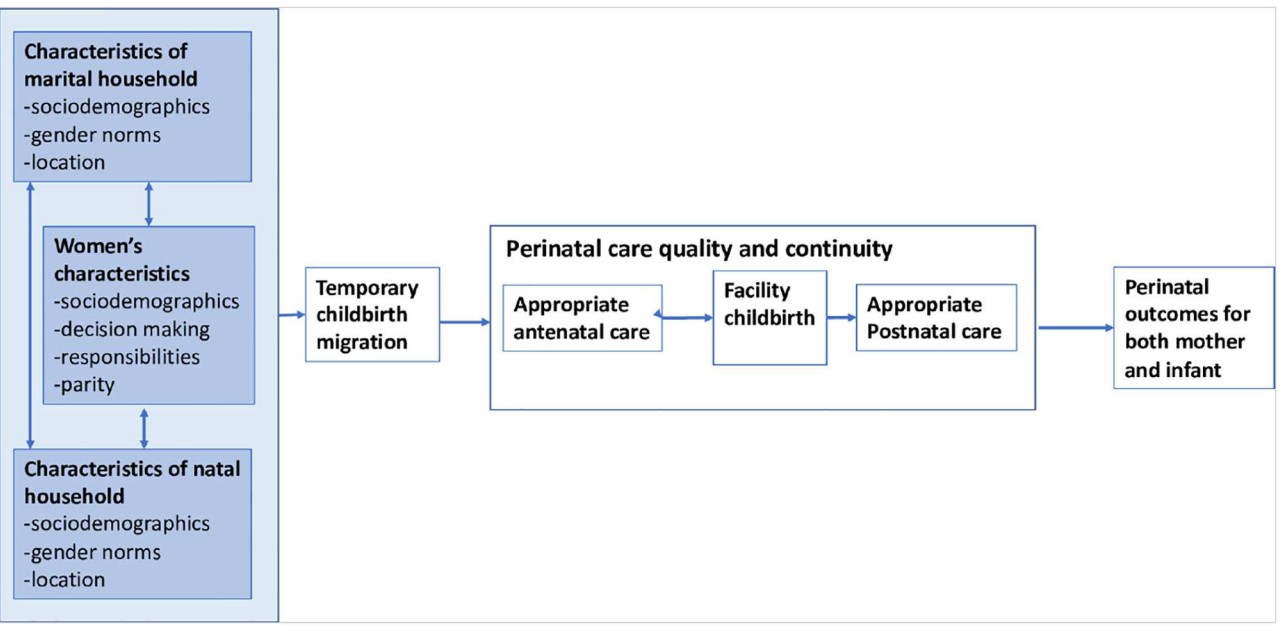

**Fig 1. Conceptual model of drivers and potential outcomes of temporary childbirth migration.**

dataset collected from representative samples of women in AWW registers in two states to quantify the impact of Temporary Childbirth Migration on health service use.

## 2. Materials and methods

Data for this study comes from a larger mixed-methods quasi-experimental evaluation of ICDS-CAS intervention for community health workers (AWWs) in the two northern Indian states of Madhya Pradesh (MP) and Bihar [51]. The evaluation aimed to assess the impact of ICDS-CAS above and beyond the business-as-usual ICDS and ISSNIP activities on service delivery of CHWs and infant and young child feeding practices [52]. These two states were selected because of the availability of comparable control districts (non-CAS ISSNIP district), willingness of the states to support the evaluation, and on suggestion of the Ministry of Woman and Child Development and the funding agency. The states were different in their health and nutrition challenges as well as the capacity of their state-funded health and nutrition systems and programs, allowing the study team to understand the rollout of ICDS-CAS in two different program environments. Madhya Pradesh and Bihar both have a high burden of under-five mortality of 69 per 1000 live births and 60 per 1000 live births respectively [5]. The stunting levels for children below the age of five are high in both the states at 44% and 49% in Madhya Pradesh and Bihar respectively. Prevalence of anaemia is 55% among pregnant women and children in both the states. Educational attainment of women in Madhya Pradesh and Bihar is quite low with only 14% and 12% of women aged 15–49 respectively having completed 12 or more years of schooling. Compared to Madhya Pradesh, the antenatal- and delivery-related indicators are, in general, worse in Bihar. 36% of mothers in Madhya Pradesh and 14% in Bihar had at least four ANC visits. 81% of women in Madhya Pradesh delivered in facilities compared to 64% in Bihar. In terms of immunization, Bihar was better than Madhya Pradesh—50% of children aged 12–23 months in Madhya Pradesh and 62% in Bihar were fully

immunised. Though they share a geographic boarder, the states are quite different in terms of governance, social development and health indictors.

## 2.1 Quantitative component

**2.1.1 Sampling and recruitment of study participants.** In this analysis, we use household level survey data collected from six administratively-defined districts in rural Madhya Pradesh and six districts of Bihar in 2019. 1200 respondents from 200 villages in each arm from each state were needed to detect a difference of 5–9 percentage point in the child health outcomes of interest in the parent study (discussed above) from the counterfactual levels between 10–50 percent with an intra-cluster correlation coefficient between 0.15–0.30 [51]. The achieved sample for endline survey (used for this analysis) consisted of 210 pairs of villages in Madhya Pradesh and 216 pairs in Bihar using nearest neighbor 1:1 propensity score matching based on variables from Census 2011 village-level characteristics from three pairs of CAS and non-CAS ISSNIP districts purposively chosen in Madhya Pradesh and Bihar. Census 2011 variables used for matching include: Distance between village and block headquarters (kilometers); Population; % of SC/ST households; % of villages served by public transport; % of villages connected to a major road; % of villages with a public ration shop; % of villages with a post office; % of villages with a bank; Average proportion of households in a village with a bank account; % of villages with an agricultural society; % of villages with a self-help group; Average proportion of households in a village serviced by closed drainage system; Average proportion of households in a village with improved source of drinking water; Average proportion of households in a village with improved sanitation facility; Average proportion of households in a village using electricity as the main source of light; Average proportion of households in a village with a pucca house; Average household asset index for the village. In the endline survey, 852 villages were selected from these 12 districts and then up to two AWCs/AWWs were sampled per village. Then we randomly sampled up to eight mothers of children <12 months and up to three pregnant women in their last trimester based on the AWW registries. More details on the evaluation design are available in our study protocol [51]. Data were collected using computer assisted personal interviews from 9,060 women in total, 4,727 from Bihar and 4,293 from Madhya Pradesh. This study excluded women not registered at Anganwadi Centers and mothers of children over 12 months and non-pregnant women. For this analysis, we further limited our analytic sample to women who were interviewed at a minimum of 6 months postpartum to ensure that our findings were not biased through selection differences by timing of migration, i.e., a greater likelihood of sample inclusion among women who remained in their marital home compared to those who had returned to their natal home. This resulted in a total analytic sample of 3,121; 1,832 in Bihar and 1,289 in Madhya Pradesh).

**2.1.2 Measures.** Sociodemographic characteristics captured included age (5-year age categories), educational attainment (illiterate/no formal education; primary (class 1–5); some secondary (class 6–8); secondary (class 9–12); more than secondary)), husband's occupation (none; agriculture; daily non-agricultural; salaried; other), woman's working status (outside home vs. not), religion (Hindu vs other), and wealth quintile. Parity was categorized as first birth vs. higher. Sociodemographic characteristics were selected a priori based on previous research in this area and published studies.

Women reported on their return to their natal home during particular time points across the perinatal period: pregnancy (0–3, 4–6, 7, 8, and 9 months), at the time of childbirth, and during the postpartum period (1, 2–5, 6–8, 9–12 months). Women who reported returning to their natal home at any time point were asked why they choose to do this by selecting all that applied from a list of options including: cultural and social norm; get better care, rest of

comfort; better or more trustworthy health care; avoid work in husband's home postpartum; to save money or because husband or family did not want to pay for childbirth; husband asked her to go; or parents asked her to come. Women self-reported on their receipt of perinatal care services relevant to where they were in the pregnancy/postpartum at the time of survey. They were asked the total number of antenatal care visits achieved, the location of childbirth (private hospital, public hospital, other, or home), and whether and how many postnatal visits they received.

We were interested in the impact of being away at the natal home at three points: the last month of pregnancy, at the time of delivery, and in the first month postpartum. For each time period we looked at one outcome related to health care visits. We selected the health care related outcomes to be reflective of guidelines and realize these may not be indicative of actual higher quality care, especially for the case of facility delivery. For pregnancy-related analyses, we made a summary variable of the number of months spent at the natal home in the last trimester of pregnancy to capture the full extent of possible antenatal care. This was operationalized as a categorical variable since there are only 4 possible values. The Indian government officially recommends a minimum of 4 ANC visits, while the WHO recently recommended 8 ANC visits [53]. At least three of these visits are supposed to be in the last month of pregnancy and two in the second to last month. Most of these higher order (4+) visits occur in the later stages of pregnancy, when women are more likely to be in their natal village. As can be seen in Fig 2 most women in India do not meet the Indian guidelines, and only 25% receive more than 4 ANC visits. Since global guidelines recommend more visits, and we know that these visits are most likely to occur at the end of pregnancy, our indicator was constructed as

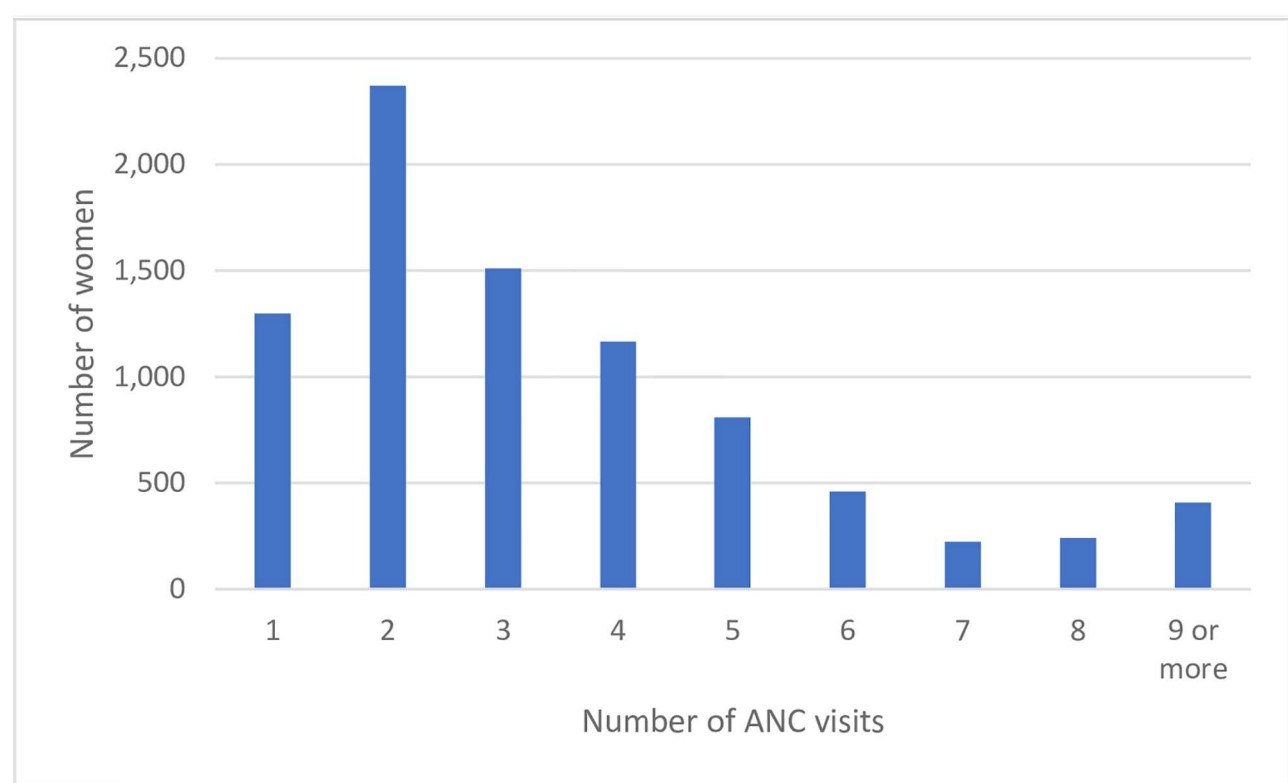

**Fig 2. Distribution of number of antenatal care visits (ANC) received by women in India, using data from the 2016 National Family Health Survey.**

continuous (number of ANC visits). For delivery-related analyses, our primary independent variable was staying at the natal home around the time of delivery. As the government incentive program has pushed the vast majority of deliveries into public facilities [54], and delivering in a private facility is seen as being higher quality, we explored the impact of being at the natal home at time of delivery on a woman delivering in a private facility versus other (public facility or home), although we include delivering in any facility versus home as a secondary analysis. For postpartum-related analyses, we operationalized adequate postpartum visits as receiving a minimum of one postpartum visit because Indian guidelines recommend one postnatal visit in the first 42 days after the mother returns to her home from delivery (post 48 hours) [53].

**2.1.3 Analysis.** *Descriptive analyses of study participants and temporary childbirth migration practices.* To characterize our study sample, we first generated descriptive statistics of participant sociodemographic characteristics overall, by state, and migration status (Table 1). We then compared the sociodemographic characteristics of our study participants by state using chi-square tests (Table 1), and due to differences identified and the contextual factors described previously, we compare state-level differences or stratify all subsequent analyses by state. We then characterized the temporary childbirth migration patterns of our study participants through describing the proportion of women who reported being at their natal home during pregnancy (any time during pregnancy, and by category: 0–3, 4–6, 7, 8, 9 months of pregnancy; Table 2), at the time of delivery, and postpartum (any time postpartum, and by category:1, 2–5, 6–8 and 9–12 months postpartum). We also tested for differences in the distribution of temporary childbirth migration and temporary childbirth migration timing by state using chi-square tests (Table 2).

*Regression modeling of sociodemographic characteristics and temporary childbirth migration.* Next we explored the relationship between participant sociodemographic characteristics and temporary childbirth migration, operationalized as returning to the natal home for at least one month during the following time points: 1) any time in the perinatal period; 2) in pregnancy; 3) for delivery; and 4) postpartum (Table 3 and S1 Table). Each of these four outcomes was modeled separately using mixed-effects logistic regression models with random effects for village to accommodate for our multi-stage sampling structure including clustering of study participants within villages and fixed effects for districts to account for time invariant district specific characteristics. Each model was specified using the following structure: $\text{logit}(p_{TCMij}) = \beta_0 + \beta_1 X_{1ij} + \beta_2 X_{2ij} + \ldots + \beta_k X_{kij} + u_{0j} + e_{0ij}$ where $p_{TCM}$ represents the probability of temporary childbirth migration at each specific time point, $\beta_0$ is our constant, $\beta_{1-k}$ represent the various sociodemographic indicators included in each model (i.e., maternal age group, maternal educational attainment, husband's occupation, whether the woman works outside the home, caste, first birth, and household wealth quintile: all selected a priori), $u_{0j}$ is the random effect specific to the *j*th village and $e_{0ij}$ is the individual-level error term for individual *i* in village *j*. Individual respondents are numbered from *i = 1 to N*, with each village (*j*).

*Descriptive analyses of reasons for temporary childbirth migration.* Next, we tabulated the reasons that women provide for returning to their natal home among the subset of women who reported any temporary childbirth migration and tested for differences in the proportion of women reporting each of the reasons by state using chi-square tests (Table 4).

*Descriptive analyses of perinatal health care outcomes.* We described receipt of community-based perinatal health care services provided by either an ASHA or AWW during temporary childbirth migration at the natal home during pregnancy (0–3, 4–6, 7, 8, 9 months of pregnancy; Table 5), at the time of delivery, and postpartum (1, 2–5, 6–8 and 9–12 months postpartum). We then described the proportion of women who achieved each of our three primary perinatal health outcomes: number of ANC visits, delivered in a private health facility, and 1 or more postnatal care visits, and one secondary perinatal health outcome: facility delivery, by

**Table 1. Sociodemographic characteristics of women 6–12 months postpartum registered at Anganwadi centers in Bihar and Madhya Pradesh, India, 2019.**

| | State | | | | Migration Status | | | | | |
| --- | --- | --- | --- | --- | --- | --- | --- | --- | --- | --- |
| | Bihar N = 1,849 | | Madhya Pradesh N = 1,296 | | Did not migrate N = 1,946 | | Migrated N = 1195 | | Total N = 3,145 | |
| | No. | % | No. | % | No. | % | No. | % | No. | % |
| Age group*** | | | | | | | | | | |
| <20 | 119 | 6.4 | 51 | 3.9 | 80 | 4.1 | 90 | 7.5 | 170 | 5.4 |
| 20–24 | 871 | 47.1 | 724 | 55.9 | 895 | 46 | 698 | 58.4 | 1,595 | 50.7 |
| 25–29 | 596 | 32.2 | 391 | 30.2 | 665 | 34.2 | 320 | 26.8 | 987 | 31.4 |
| 30+ | 263 | 14.2 | 130 | 10 | 306 | 15.7 | 87 | 7.3 | 393 | 12.5 |
| Educational attainment*** | | | | | | | | | | |
| Illiterate/no formal education | 867 | 47.2 | 415 | 32.0 | 881 | 45.5 | 400 | 33.5 | 1,282 | 40.9 |
| Primary (1–5) | 281 | 15.3 | 223 | 17.2 | 311 | 16.1 | 191 | 16 | 504 | 16.1 |
| Some secondary (6–8) | 232 | 12.6 | 324 | 25.0 | 313 | 16.2 | 243 | 20.4 | 556 | 17.7 |
| Secondary (9–12) | 382 | 20.8 | 293 | 22.6 | 376 | 19.4 | 298 | 25 | 675 | 21.5 |
| More than secondary | 76 | 4.1 | 41 | 3.2 | 55 | 2.8 | 62 | 5.2 | 117 | 3.7 |
| Occupation status of husband*** | | | | | | | | | | |
| None | 27 | 1.5 | 19 | 1.5 | 21 | 1.1 | 25 | 2.1 | 46 | 1.5 |
| Agriculture | 213 | 11.6 | 599 | 46.5 | 517 | 26.7 | 295 | 24.8 | 812 | 25.9 |
| Daily non-agricultural | 1,147 | 62.2 | 521 | 40.4 | 1,062 | 54.8 | 605 | 50.8 | 1,668 | 53.3 |
| Salaried | 394 | 21.4 | 115 | 8.9 | 281 | 14.5 | 226 | 19 | 509 | 16.3 |
| Other | 62 | 3.4 | 35 | 2.7 | 56 | 2.9 | 40 | 3.4 | 97 | 3.1 |
| Employment status*** | | | | | | | | | | |
| Works only in the home | 1,633 | 88.3 | 933 | 72.0 | 1,554 | 79.9 | 1,008 | 84.4 | 2,566 | 92.7 |
| Any work outside the home | 158 | 8.5 | 71 | 5.5 | 392 | 20.1 | 187 | 15.6 | 229 | 7.3 |
| Religion*** | | | | | | | | | | |
| Hindu | 1,691 | 91.5 | 1,225 | 94.5 | 1,829 | 94 | 1,083 | 90.6 | 2,916 | 92.7 |
| All other religions | 158 | 8.5 | 71 | 5.5 | 117 | 6 | 112 | 9.4 | 229 | 7.3 |
| Parity*** | | | | | | | | | | |
| First birth | 449 | 24.3 | 424 | 32.7 | 452 | 23.2 | 419 | 35.1 | 873 | 27.8 |
| Second or higher | 1,400 | 75.7 | 872 | 67.3 | 1,494 | 76.8 | 776 | 64.9 | 2,272 | 72.2 |
| Household wealth quintile*** | | | | | | | | | | |
| Lowest quintile | 410 | 22.2 | 246 | 19 | 458 | 23.5 | 196 | 16.4 | 656 | 20.9 |
| Lower-middle quintile | 420 | 22.7 | 207 | 16 | 420 | 21.6 | 206 | 17.2 | 627 | 19.9 |
| Middle quintile | 382 | 20.7 | 221 | 17.1 | 373 | 19.2 | 230 | 19.2 | 603 | 19.2 |
| Higher-middle quintile | 376 | 20.3 | 237 | 18.3 | 343 | 17.6 | 270 | 22.6 | 613 | 19.5 |
| Highest quintile | 261 | 14.1 | 385 | 29.7 | 352 | 18.1 | 293 | 24.5 | 646 | 20.5 |

Notes:

Results of chi-square tests assessing differences between states:

*** p<0.01,

** p<0.05, and

* p<0.1.

state (Table 6a) and migration status (Table 6b) using Wilcoxon rank sum and chi-square tests.

*Regression modeling of temporary childbirth migration and perinatal health outcomes.*
Finally, we analyzed the impact of temporary childbirth migration on each of our three primary perinatal health outcomes: number of antenatal care visits, private facility delivery, and

**Table 2. Temporal patterns in temporary childbirth migration among women 6–12 months postpartum in Bihar and Madhya Pradesh, 2019.**

| | Bihar N = 1,849 | | Madhya Pradesh N = 1,296 | | Total N = 3,145 | |
|---|---|---|---|---|---|---|
| | n | % | n | % | n | % |
| Any TCM*** | 782 | 42.4 | 413 | 31.9 | 1,195 | 38.0 |
| *Pregnancy* | | | | | | |
| Any time*** | 731 | 39.5 | 406 | 31.3 | 1,137 | 36.2 |
| Months 0–3*** | 228 | 12.3 | 95 | 7.3 | 323 | 10.3 |
| Months 4–6*** | 338 | 18.3 | 123 | 9.5 | 461 | 14.7 |
| Month 7*** | 363 | 19.6 | 163 | 12.6 | 526 | 16.7 |
| Month 8*** | 414 | 22.4 | 210 | 16.2 | 624 | 19.8 |
| Month 9*** | 499 | 27.0 | 277 | 21.4 | 776 | 24.7 |
| *At time of delivery* | | | | | | |
| Delivery*** | 504 | 27.3 | 287 | 22.1 | 791 | 25.2 |
| *Postpartum* | | | | | | |
| Any time*** | 604 | 32.7 | 316 | 24.4 | 920 | 29.3 |
| Month 1*** | 468 | 25.3 | 259 | 20.0 | 727 | 23.1 |
| Months 2–5*** | 273 | 14.8 | 145 | 11.2 | 418 | 13.3 |
| Months 6–8 | 79 | 4.3 | 48 | 3.7 | 127 | 4.0 |
| Months 9–12 | 24 | 2.7 | 14 | 2.6 | 38 | 2.7 |

Notes:

TCM: temporary childbirth migration.

Results of chi-square tests assessing differences between states:

*** p<0.01,

** p<0.05, and

* p<0.1.

postpartum health check (Table 7), and one secondary perinatal health outcome: facility delivery (S2 Table), stratified by state. Each of these outcomes was modeled separately using mixed effects generalized structural equation models [55] to simultaneously model the perinatal health outcome and temporary childbirth migration. We selected this approach to accommodate the potential for endogeneity issues whereby certain unobserved factors (e.g., socioeconomic status, health status, etc.) might jointly influence both the likelihood of temporary childbirth migration and the perinatal health outcome, introducing bias into our estimation of this key relationship of interest. To reduce the risk of endogeneity bias, we allowed the error terms in the two equations to be correlated to monitor and mitigate selection bias in those that chose temporary migration. These models accommodated both our multi-stage sampling structure (i.e., through including district as a fixed effect and incorporating random effects at the village level) and our need to combine different model types (i.e., Poisson and logistic) for our varied dependent variable structure [56]. To develop our final models, we first estimated preliminary models evaluating the relationship between sociodemographic characteristics (i.e., maternal age group, maternal educational attainment, husband's occupation, whether the woman works outside the home, caste, first birth, and household wealth quintile) and each temporary childbirth migration variable of interest to identify which characteristics were conservatively (p<0.1) associated. We then included only the sociodemographic characteristics which met this threshold of statistical significance within each final selection model. Final analyses for the different outcomes followed a similar strategy but differed by dependent

**Table 3. Odds of temporary childbirth migration by sociodemographic characteristics among women 6–12 months postpartum, Bihar and Madhya Pradesh, 2019.**

| | Anytime during the perinatal period | | During pregnancy | | For delivery | | Postpartum | |
|---|---|---|---|---|---|---|---|---|
| | Bihar | Madhya Pradesh | Bihar | Madhya Pradesh | Bihar | Madhya Pradesh | Bihar | Madhya Pradesh |
| | N = 1,828 | N = 1,289 | N = 1,832 | N = 1,289 | N = 1,832 | N = 1,289 | N = 1,832 | N = 1,289 |
| | OR | OR | OR | OR | OR | OR | OR | OR |
| | (95% CI) | (95% CI) | (95% CI) | (95% CI) | (95% CI) | (95% CI) | (95% CI) | (95% CI) |
| Age group (compared to <20) | | | | | | | | |
| 20–24 | 0.84 | 0.83 | 0.68 | 1.31 | 0.77 | 0.89 | 0.83 | 1.04 |
| | (0.55–1.27) | (0.44–1.57) | (0.45–1.03) | (0.68–2.53) | (0.50–1.17) | (0.45–1.79) | (0.55–1.26) | (0.52–2.06) |
| 25–29 | 0.58* | 0.55 | 0.54** | 0.98 | 0.58* | 0.53 | 0.62* | 0.58 |
| | (0.37–0.92) | (0.27–1.10) | (0.34–0.85) | (0.48–2.00) | (0.36–0.93) | (0.25–1.15) | (0.39–0.99) | (0.27–1.24) |
| 30+ | 0.38*** | 0.29** | 0.31*** | 0.36* | 0.35*** | 0.38* | 0.42** | 0.37* |
| | (0.22–0.63) | (0.13–0.68) | (0.18–0.53) | (0.15–0.87) | (0.20–0.63) | (0.15–0.97) | (0.24–0.72) | (0.15–0.93) |
| Women's education group (compared to illiterate/no formal education) | | | | | | | | |
| Primary (1–5) | 1.09 | 1.50 | 1.30 | 1.33 | 1.20 | 1.16 | 1.20 | 1.10 |
| | (0.82–1.46) | (0.99–2.27) | (0.97–1.75) | (0.88–2.02) | (0.87–1.65) | (0.73–1.85) | (0.89–1.63) | (0.70–1.72) |
| Some secondary (6–8) | 1.29 | 1.74** | 1.33 | 1.51* | 1.13 | 1.33 | 1.43* | 1.46 |
| | (0.94–1.77) | (1.17–2.59) | (0.96–1.82) | (1.01–2.25) | (0.79–1.59) | (0.85–2.08) | (1.03–1.98) | (0.95–2.24) |
| Secondary (9–12) | 1.13 | 1.74* | 1.32 | 1.45 | 1.09 | 1.47 | 1.20 | 1.42 |
| | (0.85–1.51) | (1.14–2.65) | (0.99–1.76) | (0.95–2.22) | (0.80–1.50) | (0.91–2.38) | (0.89–1.61) | (0.90–2.25) |
| More than secondary | 1.34 | 3.29** | 1.33 | 3.65** | 1.47 | 3.74** | 2.00* | 2.49* |
| | (0.78–2.28) | (1.48–7.30) | (0.78–2.26) | (1.59–8.37) | (0.85–2.55) | (1.53–9.15) | (1.17–3.42) | (1.06–5.84) |
| Husband's occupation (compared to daily non-agricultural) | | | | | | | | |
| Agriculture | 1.06 | 1.40* | 1.01 | 1.53** | 0.93 | 1.28 | 0.81 | 1.09 |
| | (0.78–1.44) | (1.04–1.87) | (0.74–1.38) | (1.14–2.06) | (0.66–1.31) | (0.92–1.78) | (0.58–1.13) | (0.80–1.49) |
| Salaried | 1.07 | 1.45 | 1.02 | 0.97 | 0.88 | 0.91 | 1.14 | 1.09 |
| | (0.84–1.38) | (0.89–2.35) | (0.79–1.31) | (0.58–1.63) | (0.67–1.16) | (0.51–1.63) | (0.88–1.47) | (0.64–1.85) |
| Other | 1.14 | 1.11 | 1.28 | 1.17 | 0.70 | 0.76 | 0.70 | 0.74 |
| | (0.66–1.96) | (0.51–2.43) | (0.75–2.19) | (0.53–2.59) | (0.37–1.32) | (0.31–1.85) | (0.39–1.28) | (0.32–1.74) |
| None | 2.71* | 1.24 | 1.33 | 1.14 | 0.66 | 1.07 | 1.39 | 0.88 |
| | (1.16–6.35) | (0.45–3.42) | (0.58–3.04) | (0.40–3.20) | (0.25–1.75) | (0.36–3.18) | (0.60–3.21) | (0.30–2.56) |
| Woman works outside the home | 0.71* | 1.31 | 0.63** | 1.70*** | 0.58** | 1.70** | 0.65* | 1.22 |
| | (0.51–0.98) | (0.97–1.77) | (0.45–0.88) | (1.26–2.31) | (0.39–0.85) | (1.22–2.38) | (0.45–0.92) | (0.88–1.69) |
| Hindu (compared to other) | 0.62** | 0.77 | 0.95 | 0.51* | 0.94 | 0.51* | 1.10 | 0.51* |
| | (0.43–0.88) | (0.45–1.31) | (0.66–1.36) | (0.30–0.88) | (0.63–1.41) | (0.29–0.88) | (0.75–1.62) | (0.30–0.88) |
| First birth (compared to second or higher) | 1.27 | 1.36* | 1.45** | 1.62** | 1.30 | 1.40 | 1.28 | 1.13 |
| | (0.98–1.65) | (1.01–1.85) | (1.12–1.87) | (1.19–2.20) | (0.99–1.72) | (1.00–1.97) | (0.99–1.67) | (0.82–1.57) |
| Wealth quintile (compared to poorest) | | | | | | | | |
| Poor | 1.10 | 0.86 | 0.95 | 1.09 | 0.83 | 1.77* | 0.93 | 1.38 |
| | (0.82–1.49) | (0.53–1.38) | (0.70–1.28) | (0.68–1.74) | (0.60–1.16) | (1.04–3.01) | (0.68–1.27) | (0.82–2.32) |
| Middle | 1.37* | 1.03 | 1.07 | 0.99 | 0.99 | 1.36 | 0.92 | 1.38 |
| | (1.01–1.86) | (0.65–1.63) | (0.79–1.46) | (0.62–1.58) | (0.71–1.39) | (0.79–2.34) | (0.66–1.27) | (0.82–2.32) |
| Wealthy | 1.60** | 1.33 | 1.15 | 1.37 | 1.06 | 1.56 | 1.18 | 1.39 |
| | (1.16–2.20) | (0.85–2.09) | (0.83–1.58) | (0.87–2.17) | (0.75–1.51) | (0.91–2.66) | (0.85–1.65) | (0.83–2.32) |
| Wealthiest | 1.47* | 1.26 | 1.00 | 1.14 | 1.36 | 1.47 | 1.35 | 1.55 |
| | (1.01–2.13) | (0.81–1.95) | (0.69–1.46) | (0.73–1.78) | (0.92–2.02) | (0.88–2.45) | (0.93–1.98) | (0.95–2.53) |

*(Continued)*

**Table 3.** (Continued)

| | Anytime during the perinatal period | | During pregnancy | | For delivery | | Postpartum | |
|---|---|---|---|---|---|---|---|---|
| | Bihar | Madhya Pradesh | Bihar | Madhya Pradesh | Bihar | Madhya Pradesh | Bihar | Madhya Pradesh |
| | N = 1,828 | N = 1,289 | N = 1,832 | N = 1,289 | N = 1,832 | N = 1,289 | N = 1,832 | N = 1,289 |
| | OR | OR | OR | OR | OR | OR | OR | OR |
| | (95% CI) | (95% CI) | (95% CI) | (95% CI) | (95% CI) | (95% CI) | (95% CI) | (95% CI) |
| District | | | | | | | | |
| 1 | 1.49* | 1.02 | 1.19 | 0.98 | 1.27 | 0.81 | 0.91 | 0.93 |
| | (1.05–2.11) | (0.69–1.50) | (0.83–1.70) | (0.67–1.45) | (0.85–1.88) | (0.55–1.21) | (0.63–1.31) | (0.63–1.37) |
| 2 | 1.50* | 0.86 | 1.45* | 0.68 | 1.19 | 0.68 | 1.14 | 0.54** |
| | (1.06–2.12) | (0.57–1.30) | (1.02–2.05) | (0.45–1.03) | (0.80–1.76) | (0.44–1.05) | (0.80–1.62) | (0.35–0.83) |
| 3 | 2.12*** | 0.29*** | 1.67** | 0.24*** | 1.74** | 0.12*** | 1.49* | 0.16*** |
| | (1.49–3.03) | (0.18–0.48) | (1.17–2.38) | (0.14–0.39) | (1.18–2.58) | (0.060–0.24) | (1.04–2.14) | (0.087–0.29) |
| 4 | 2.50*** | 0.19*** | 2.46*** | 0.13*** | 2.52*** | 0.057*** | 2.04*** | 0.12*** |
| | (1.76–3.55) | (0.12–0.31) | (1.73–3.49) | (0.079–0.22) | (1.72–3.69) | (0.026–0.12) | (1.43–2.91) | (0.069–0.22) |
| 5 | 2.05*** | 0.25*** | 1.70** | 0.25*** | 1.59* | 0.18*** | 1.04 | 0.25*** |
| | (1.45–2.89) | (0.15–0.41) | (1.20–2.39) | (0.15–0.40) | (1.08–2.33) | (0.10–0.32) | (0.73–1.49) | (0.15–0.42) |

*** p<0.001, ** p<0.01, * p<0.05.

Notes:

CI: Confidence intervals; OR: Odds ratios

Temporary childbirth migration is defined as being away from the marital home for at least one month during the perinatal period (or specified component of the perinatal period).

P-values for categorical comparison of odds of index category with reference category: *** p<0.01, ** p<0.05, * p<0.1

**Table 4. Reasons for temporary childbirth migration among women 6–12 months postpartum who reported returning to their natal home at any point during the perinatal period, Bihar and Madhya Pradesh, 2019.**

| Reason for temporary childbirth migration | Bihar | | Madhya Pradesh | | Total | |
|---|---|---|---|---|---|---|
| | N = 782 | | N = 413 | | N = 1,195 | |
| | N | % | N | % | N | % |
| It is the cultural and social norm *** | 67 | 8.6 | 133 | 32.2 | 200 | 16.7 |
| Will get better care, rest or comfort | 438 | 56.0 | 219 | 53.0 | 657 | 55.0 |
| There is better or more trustworthy health care there | 187 | 23.9 | 84 | 20.3 | 271 | 22.7 |
| Avoid work in the husband's home postpartum | 69 | 8.8 | 44 | 10.7 | 113 | 9.5 |
| To save money or because husband/ husband's family didn't want to pay for childbirth | 31 | 4.0 | 20 | 4.8 | 51 | 4.3 |
| Husband's family asked me to go | 91 | 11.6 | 43 | 10.4 | 134 | 11.2 |
| Parents asked me to come *** | 157 | 20.1 | 115 | 27.8 | 272 | 22.8 |

Notes:

Results of chi-square tests assessing differences between states:

*** p<0.01,

** p<0.05, and

* p<0.1.

Respondents were allowed to select more than one response; therefore, columns will not total 100%.

**Table 5. Service receipt from AWW and/or ASHA while at natal home by temporary childbirth migration timing among pregnant and postpartum women in Bihar and Madhya Pradesh, 2019.**

| | Received services from AWW and/or ASHA while at natal home ^ | | |
|---|---|---|---|
| | **Bihar** | **Madhya Pradesh** | **Total** |
| | **n (%)** | **n (%)** | **n (%)** |
| *Pregnancy* | | | |
| Months 0–3 | 62 (27.4) | 59 (62.1) | 121 (37.7) *** |
| Months 4–6 | 106 (31.5) | 65 (52.9) | 171 (37.3) *** |
| Month 7 | 104 (28.8) | 97 (59.5) | 201 (38.4) *** |
| Month 8 | 105 (25.4) | 120 (57.1) | 225 (36.1) *** |
| Month 9 | 135 (27.1) | 150 (54.2) | 285 (36.8) *** |
| *Delivery* | | | |
| At time of delivery | 176 (35.0) | 155 (54.0) | 331 (41.9) *** |
| *Postpartum* | | | |
| Month 1 | 148 (31.7) | 149 (57.5) | 297 (40.9) *** |
| Months 2–5 | 93 (34.2) | 92 (63.5) | 185 (44.4) *** |
| Months 6–8 | 20 (25.3) | 30 (62.5) | 50 (39.4) *** |
| Months 9–12 | 4 (18.2) | 9 (64.3) | 13 (36.1) *** |

Notes:

Results of chi-square tests assessing differences between states:

*** p<0.01,

** p<0.05, and

* p<0.1.

^ The denominator changes for each time period because only women who were at their natal home within that period of pregnancy/postpartum were asked if they availed services at that place.

variable structure. For example, analyses of temporary childbirth migration dose (i.e., number of months spent in natal home during the third trimester of pregnancy) on the number of ANC visits achieved used Poisson regression models following the two-model structure: 1) $\log(\lambda_{TCMij}) = \beta_0 + \beta_1 X_{1ij} + \beta_2 X_{2ij} + \ldots + \beta_k X_{kij} + u_{0j} + e_{0ij}$ where $\lambda_{TCMij}$ represents temporary childbirth migration dose, $\beta_0$ is our constant, $\beta_{1-k}$ represent the various sociodemographic indicators included in each model, $u_{0j}$ is the random effect specific to the $j$th village and $e_{0ij}$ is the individual-level error term for individual $i$; and 2) $\log(\lambda_{ANCij}) = \beta_0 + \beta_1 X_{1ij} + \beta_2 X_{2ij} + \ldots + \beta_k X_{kij} + u_{0j} + e_{0ij}$ where $\lambda_{ANCij}$ represents the number of antenatal care visits, $\beta_1$ represents the temporary childbirth migration dose, $\beta_{2-k}$ represent the various sociodemographic indicators included and fixed effects for each district, $u_{0j}$ is the random effect specific to the $j$th village, and $e_{0ij}$ is the individual-level error term for individual $i$. Impact of temporary childbirth migration (i.e., natal home at time of childbirth for delivery analyses and natal home during the first month postpartum for postpartum analyses, respectively) on our two delivery-related outcomes (private versus other facility birth, and facility versus home birth) and achievement of one or more postnatal care visits (PNC) employed logistic regression, following the structure: 1) $\text{logit}(p_{TCMij}) = \beta_0 + \beta_1 X_{1ij} + \ldots + \beta_k X_{kij} + u_{0j} + e_{0ij}$ where p represents temporary childbirth migration, $\beta_{1-k}$ represent the various sociodemographic indicators included and fixed effects for each district, $u_{0j}$ is the random effect specific to the $j$th village, and $e_{0ij}$ is the individual-level error term for individual $i$; and 2) $\text{logit}(p_{DELVij}) = \beta_0 + \beta_1 X_{1ij} + \beta_2 X_{2ij} + \ldots + \beta_k X_{kij} + u_{0j} + e_{0ij}$ where p represents the probability of the delivery-related outcome, $\beta_1$ represents temporary childbirth migration, $\beta_{2-k}$ represent the various sociodemographic indicators included and district fixed effects, $u_{0j}$ is the random effect specific to the $j$th village, and $e_{0ij}$ is

**Table 6.** a: Perinatal care achievement among women 6–12 months postpartum by state, 2019. b: Perinatal care achievement among women 6–12 months postpartum by temporary childbirth migration status, total and by state, 2019.

| Perinatal care outcome | Bihar | | Madhya Pradesh | | Total | |
|---|---|---|---|---|---|---|
| | N = 1,849 | | N = 1,296 | | N = 3,145 | |
| | Median | IQR | Median | IQR | Median | IQR |
| Number of ANC visits*** | 2 | 1–3 | 4 | 2–6 | 3 | 2–4 |
| | N | % | N | % | N | % |
| Delivered in a private facility*** | 278 | 15.0 | 96 | 7.4 | 374 | 11.9 |
| Received 1 or more postnatal care visit | 1071 | 48.1 | 421 | 45.8 | 149 | 47.4 |
| Perinatal care outcome | Bihar | | Madhya Pradesh | | Total | |
| | N = 1,849 | | N = 1,296 | | N = 3,145 | |
| | Median | IQR | Median | IQR | Median | IQR |
| Number of ANC visits | | | | | | |
| Migrated during pregnancy | 2*** | 2–3 | 4*** | 3–6 | 3** | 2–5 |
| Did not migrate | 2 | 1–3 | 4 | 3–5 | 3 | 2–4 |
| | N | % | N | % | N | % |
| Delivered in a private facility | | | | | | |
| Migrated for delivery | 92** | 18.3 | 33** | 11.5 | 125*** | 15.8 |
| Did not migrate | 186 | 13.8 | 63 | 6.2 | 249 | 10.6 |
| Received 1 or more postnatal care visits | | | | | | |
| Migrated postpartum | 45 | 40.9 | 303 | 44.5 | 348 | 44.0 |
| Did not migrate | 1026 | 48.5 | 118 | 49.4 | 1144 | 48.6 |

Notes:

ANC: antenatal care; IQR: interquartile range

Results of chi-square tests assessing differences between states (in perinatal care outcome column):

*** p<0.01,

** p<0.05, and

* p<0.1.

the individual-level error term for individual *i*. For each model, we monitored for evidence of selection and for models where selection was not identified (i.e., individual random effect variation was 0), we excluded this parameter from the final model presented. No selection was identified within two models: 1) for the outcome of any facility delivery for Bihar (supplementary) and 2) for the outcome of private facility delivery for Madhya Pradesh. We conducted sensitivity analyses modeling district as a fixed-effect versus mixed-effects (Table 7). All analyses were conducted using Stata v.16 [57], and analysis and data files are publicly accessible on GitHub (https://git.ucsf.edu/alison-elayadi/temporary-childbirth-migration.git).

## 2.2 Qualitative component

A qualitative study was conducted in one district in each state in March-April 2018. A total of 32 AWWs and 55 women were interviewed by a team of four Indian female researchers (including the second author (LG)). In Madhya Pradesh, we interviewed 17 AWWs, 13 pregnant women and 17 postpartum women. In Bihar, we interviewed 15 AWWs, 12 pregnant women and 13 postpartum women. The interview guides covered topics related to the intervention in the larger study, as well as questions on knowledge and behaviors related to pregnancy and postpartum infant and maternal health care, and AWW-woman interactions. Interviews were conducted until saturation was reached. Interviews were audio-recorded with

**Table 7. Relationship between temporary childbirth migration during the perinatal period and receipt of perinatal care across three perinatal care periods among women 6–12 months postpartum, Bihar and Madhya Pradesh, 2019.**

| | Number of ANC Visits | | Private Facility Delivery | | Postpartum Health Check | |
|---|---|---|---|---|---|---|
| | Bihar (N = 1,838) | MP (N = 1,289) | Bihar (N = 1,832) | MP (N = 1,277) | Bihar (N = 1,832) | MP (N = 1,289) |
| | IRR (95% CI) | IRR (95% CI) | OR (95% CI) | OR (95% CI) | OR (95% CI) | OR (95% CI) |
| Temporary Childbirth Migration | | | | | | |
| Months at Natal Home, 3rd Tri | 1.05** (1.02–1.08) | 0.98 (0.92–1.04) | | | | |
| Natal Home at Childbirth | | | 0.07 (0.00–20.60) | 0.89 (0.41–1.92) | | |
| Natal Home Postpartum | | | | | 1.60 (0.23–11.0) | 3.63 (0.13–97.9) |
| Age Group | | | | | | |
| <20 | | | | REF | | |
| 20–24 | | | | 1.25 (0.26–6.04) | | |
| 25–29 | | | | 2.63 (0.53–13.0) | | |
| 30+ | | | | 5.26* (1.01–27.4) | | |
| Educational Attainment | | | | | | |
| None | REF | REF | REF | REF | REF | REF |
| Primary (1–5) | 1.13* (1.02–1.24) | 1.16** (1.05–1.28) | 4.22 (0.81–22.2) | 1.74 (0.58–5.28) | 1.11 (0.80–1.53) | 1.68 (0.94–2.98) |
| Some secondary (6–8) | 1.18*** (1.07–1.31) | 1.14** (1.05–1.23) | 0.84 (0.19–3.74) | 6.27*** (2.23–17.60) | 1.30 (0.89–1.89) | 1.29 (0.77–2.18) |
| Secondary (9–12) | 1.17** (1.06–1.29) | 1.12* (1.02–1.23) | 6.29* (1.29–30.8) | 10.10*** (3.43–29.90) | 1.42 (0.98–2.05) | 1.30 (0.78–2.15) |
| More than secondary | 1.29** (1.08–1.54) | 1.38** (1.14–1.67) | 10.2 (0.80–130) | 50.45*** (13.85–183.76) | 1.58 (0.77–3.26) | 1.52 (0.50–4.60) |
| Husband Occupation | | | | | | |
| Daily non-agricultural | REF | REF | | | | REF |
| Agriculture | 1.06 (0.95–1.18) | 1.09* (1.02–1.16) | | 1.87* (1.04–3.35) | | 0.99 (0.68–1.44) |
| Salaried | 1.06 (0.97–1.16) | 1.19** (1.07–1.33) | | 2.15* (1.00–4.62) | | 1.82 (0.84–3.99) |
| Other | 0.71*** (0.63–0.81) | 0.87 (0.70–1.08) | | 0.64 (0.10–3.99) | | 0.13* (0.02–0.85) |
| None | 0.89 (0.69–1.15) | 1.01 (0.72–1.39) | | - | | 0.66 (0.21–2.11) |
| Respondent Works Out of Home | | | | | 1.99** (1.30–3.03) | |
| Hindu vs. Other Caste | | 0.87* (0.76–0.99) | | | | |
| First Birth vs. Higher | 1.17*** (1.08–1.27) | 1.06 (1.00–1.13) | 6.52** (1.99–21.3) | | | |
| Household Wealth Quintile | | | | | | |
| Lowest quintile | REF | REF | REF | | REF | |
| Lower-middle quintile | 1.09 (0.99–1.19) | 1.01 (0.91–1.11) | 1.46 (0.31–6.85) | | 1.16 (0.84–1.59) | |
| Middle quintile | 1.09 (0.99–1.20) | 1.07 (0.97–1.19) | 13.3* (1.48–119) | | 1.19 (0.87–1.63) | |
| Higher-middle quintile | 1.29*** (1.17–1.43) | 1.15** (1.05–1.27) | 14.8* (1.53–143) | | 1.17 (0.82–1.68) | |
| Highest quintile | 1.34*** (1.20–1.51) | 1.13* (1.02–1.25) | 78.3** (3.23–1,899) | | 1.58* (1.05–2.39) | |
| District | | | | | | |
| 1 | REF | REF | REF | REF | REF | REF |
| 2 | 1.22** (1.08–1.38) | 1.21*** (1.10–1.33) | 0.44 (0.063–3.12) | 0.33** (0.14–0.75) | 0.54** (0.35–0.82) | 3.69* (1.21–11.3) |
| 3 | 1.04 (0.92–1.19) | 0.68*** (0.61–0.77) | 1.80 (0.39–8.35) | 0.06*** (0.02–0.22) | 0.81 (0.54–1.23) | 0.21** (0.08–0.57) |
| 4 | 1.04 (0.92–1.17) | 0.85** (0.77–0.95) | 0.30 (0.025–3.68) | 0.09*** (0.03–0.24) | 0.73 (0.46–1.14) | 0.92 (0.31–2.71) |
| 5 | 1.05 (0.93–1.18) | 0.95 (0.86–1.06) | 0.75 (0.068–8.25) | 0.13*** (0.05–0.35) | 0.90 (0.54–1.48) | 6.63 (0.62–71.1) |
| 6 | 1.37*** (1.23–1.53) | 0.63*** (0.56–0.72) | 1.94 (0.35–10.6) | 0.15*** (0.05–0.46) | 0.98 (0.65–1.48) | 0.75 (0.28–1.98) |
| Random parameters | | | | | | |
| M1[village] [a] | 1 (const) | 1 (const) | 1 (const) | 1 (const) | 1 (const) | 1 (const) |
| L [individual] [a] | 735.49 (1.36.86) | 0.16 (0.13) | 6.04 (8.19) | - | -0.09 (0.20) | 0.20 (0.50) |
| **Temporary Childbirth Migration Variable** | **Months in natal home, 3rd trimester** | | **Natal home for childbirth** | | **Natal home postpartum** | |
| Maternal Age Group | | | | | | |
| <20 | REF | | REF | | REF | REF |

*(Continued)*

**Table 7.** (Continued)

| | Number of ANC Visits | | Private Facility Delivery | | Postpartum Health Check | |
|---|---|---|---|---|---|---|
| | Bihar (N = 1,838) | MP (N = 1,289) | Bihar (N = 1,832) | MP (N = 1,277) | Bihar (N = 1,832) | MP (N = 1,289) |
| | IRR (95% CI) | IRR (95% CI) | OR (95% CI) | OR (95% CI) | OR (95% CI) | OR (95% CI) |
| 20–24 | 0.74 (0.48–1.15) | | 0.76 (0.47–1.24) | 0.88 (0.44–1.77) | 0.58 (0.20–1.70) | 0.91 (0.27–3.07) |
| 25–29 | 0.65 (0.40–1.05) | | 0.54 (0.27–1.06) | 0.55 (0.26–1.17) | 0.31 (0.09–1.10) | 0.30 (0.024–3.71) |
| 30+ | 0.35*** (0.20–0.64) | | 0.29** (0.12–0.69) | 0.37* (0.14–0.99) | 0.12** (0.03–0.59) | 0.12 (0.0025–5.99) |
| Maternal Educational Attainment | | | | | | |
| None | REF | REF | | | | |
| Primary (1–5) | 1.47* (1.09–1.97) | 1.25 (0.75–2.09) | | 1.19 (0.74–1.92) | 1.65 (0.79–3.43) | 1.06 (0.47–2.41) |
| Some secondary (6–8) | 1.27 (0.92–1.77) | 1.44 (0.88–2.34) | | 1.43 (0.92–2.23) | 2.71* (1.11–6.58) | 2.12 (0.42–10.6) |
| Secondary (9–12) | 1.25 (0.94–1.65) | 1.58 (0.90–2.77) | | 1.56 (0.93–2.62) | 2.15 (1.00–4.62) | 2.01 (0.44–9.05) |
| More than secondary | 1.57 (0.97–2.54) | 4.96** (1.74–14.1) | | 3.77** (1.50–9.44) | 9.46* (1.69–52.8) | 6.02 (0.19–195) |
| Husband Occupation | | | | | | |
| Daily non-agricultural | | REF | | | | |
| Agriculture | | 1.34 (0.93–1.93) | | | | |
| Salaried | | 0.84 (0.45–1.58) | | | | |
| Other | | 0.79 (0.27–2.32) | | | | |
| None | | 0.81 (0.26–2.56) | | | | |
| Respondent Works Outside Home | 0.59** (0.39–0.88) | 0.32** (0.15–0.71) | 0.52** (0.34–0.80) | 1.79*** (1.28–2.50) | 0.32* (0.12–0.89) | |
| Hindu vs. Other Caste | | 1.84** (1.23–2.75) | | 0.52* (0.29–0.94) | | 0.26 (0.02–4.11) |
| First Birth vs. Higher | 1.50** (1.14–1.99) | 1.72*** (1.26–2.35) | 1.43* (1.02–2.00) | 1.43 (1.00–2.04) | 2.06 (0.93–4.54) | |
| Household Wealth Quintile | | | | | | |
| Lowest quintile | | REF | | REF | | REF |
| Lower-middle quintile | | 1.91* (1.01–3.61) | | 1.92* (1.11–3.33) | | 2.20 (0.32–15.1) |
| Middle quintile | | 1.43 (0.76–2.66) | | 1.28 (0.73–2.26) | | 1.98 (0.33–11.9) |
| Higher-middle quintile | | 1.68 (0.91–3.12) | | 1.48 (0.87–2.51) | | 1.96 (0.44–8.79) |
| Highest quintile | | 1.67 (0.95–2.92) | | 1.40 (0.85–2.30) | | 2.55 (0.33–19.8) |
| District | | | | | | |
| 1 | REF | REF | REF | | REF | REF |
| 2 | 1.38 (0.93–2.05) | 0.82 (0.50–1.33) | 1.41 (0.88–2.27) | 0.84 (0.55–1.28) | 0.77 (0.31–1.89) | 0.89 (0.38–2.10) |
| 3 | 1.40 (0.94–2.08) | 0.69 (0.41–1.13) | 1.32 (0.83–2.10) | 0.71 (0.46–1.10) | 1.29 (0.55–3.00) | 0.34 (0.034–3.39) |
| 4 | 1.56* (1.05–2.32) | 0.088*** (0.038–0.20) | 2.08** (1.22–3.56) | 0.12*** (0.06–0.26) | 2.53 (1.00–6.43) | 0.03 (0.00–19.40) |
| 5 | 2.58*** (1.76–3.79) | 0.081*** (0.035–0.19) | 3.13*** (1.62–6.04) | 0.063*** (0.03–0.14) | 5.49** (1.63–18.40) | 0.02 (0.00–33.6) |
| 6 | 1.55* (1.05–2.27) | 0.19*** (0.096–0.39) | 1.87* (1.09–3.18) | 0.19*** (0.10–0.35) | 1.12 (0.47–2.66) | 0.09 (0.00–10.50) |
| Random parameters | | | | | | |
| M1[village][a] | -0.50 (0.73) | 2.20 (1.20) | -0.12 (0.12) | 0.29 (0.27) | -0.59 (1.15) | -0.02 (0.21) |
| L[individual] [a] | 1 (const) | 1 (const) | 1 (const) | - | 1 (const) | 1 (const) |
| Random parameters | | | | | | |
| var(M1[village][b] | 0.036 (0.009) | 0.02 (0.01) | 4.59 (4.91) | 1.08 (0.63) | 0.28 (0.12) | 2.04 (1.49) |

*(Continued)*

**Table 7.** (Continued)

| | Number of ANC Visits | | Private Facility Delivery | | Postpartum Health Check | |
|---|---|---|---|---|---|---|
| | Bihar (N = 1,838) | MP (N = 1,289) | Bihar (N = 1,832) | MP (N = 1,277) | Bihar (N = 1,832) | MP (N = 1,289) |
| | IRR (95% CI) | IRR (95% CI) | OR (95% CI) | OR (95% CI) | OR (95% CI) | OR (95% CI) |
| var(L) [individual] [b] | 0.000 (0.000) | 0.66 (0.97) | 0.72 (1.61) | - | 12.27 (6.90) | 7.98 (21.20) |

Notes:

IRR: Incidence rate ratio; OR: Odds ratio

[a] β(SE),

[b]Var(SE)

Analysis represents joint modeling of 1) sociodemographic characteristics on temporary childbirth migration (bottom section of table) and 2) temporary childbirth migration on perinatal care receipt during three perinatal care phases: antenatal, delivery, and postpartum (top section of table).

Outcomes in count format were modeled using Poisson regression models whereas outcomes in binary format were modeled using logistic regression models.

Mixed effects models were used to accommodate clustering at the village level due to sampling and to provide a population-averaged estimate. District effects are modeled as fixed-effects to explicitly account for time invariant district specific characteristics.

Where no selection was identified, this parameter was removed from the final model and is noted in the table above by (–) within the var (L) row.

Results of joint modeling for outcome any facility delivery are in S1 Table.

permission. The average length of interviews was 90 minutes for AWWs and 30 minutes for women. Interviews were conducted in a private setting either at the homes of the beneficiaries or at the Anganwadi centers for the AWWs.

**2.2.1 Analysis.** All the audio-recorded interviews were anonymized, transcribed verbatim and translated into English. We followed the steps involved in thematic analysis, including familiarizing ourselves with the data, generating initial codes, searching, reviewing, and finally defining themes. A team of four female researchers experienced in qualitative research (two Indian, two American, including all three authors) reviewed and coded the transcripts in Dedoose [58]. Additional codes were developed as identified in an iterative manner. At least 10% of the AWW and beneficiary transcripts were double coded by all the researchers to ensure inter-rater reliability in coding. Thematic analysis was conducted by three of the authors (NDS, LG, AE) and an Indian research assistant to develop themes based on the code reports generated. Disagreements were discussed and resolved with all team members involved [59].

### 2.3 Ethics

Study protocols were reviewed and approved by institutional review boards at the University of California, Berkeley (Ref. No. 2016-08-9092), and the India-based Suraksha Independent Ethics Committee (Protocol No. 2016-08-9092). All participants provided informed consent. The interviewer read the informed consent, then the respondent signed on the tablet and was asked to read a sentence providing verbal consent, which was audio recorded on the tablet ("I have understood the purpose of the survey and my rights as the respondent. I agree to participate in this interview.") The trial is registered at https://doi.org/10.1186/ISRCTN83902145.

## 3. Results

### 3.1 Study sample

Sociodemographic characteristics of our study sample and differences in sociodemographic characteristics by state and by migration status are presented in Table 1. Most study participants were aged 20–24 (47.1% Bihar, 55.9% Madhya Pradesh, no formal education (47.2%

Bihar and 32.0% Madhya Pradesh) and works only in the home (88.3% Bihar and 72.0% Madhya Pradesh). The index pregnancy/childbirth was the first birth for 24.3% of participants in Bihar and 32.7% in Madhya Pradesh. All sociodemographic characteristics differed significantly between states and between migration status.

## 3.2 When do women return to their natal home during the perinatal period?

The distribution of temporary childbirth migration patterns is described across the perinatal period in Table 2. Nearly two-fifths of all women reported having spent at least one month of their most recent perinatal period at their natal home (38.0%). The distribution of when women migrated to their natal homes is bell-shaped, with the largest proportion (25.2%) away for delivery, and slightly fewer (24.7%) away in the 9th month of pregnancy or during the first month postpartum (23.1%). About a tenth (10.3%) of women report migrating to their natal home in early pregnancy (month 0–3), with steady increases until delivery. Most women report returning to their marital home after one month postpartum, with only 13.3% remaining through months 2–5. More women were away from their marital home in Bihar compared to Madhya Pradesh in most time windows, with no differences after 6 months postpartum.

## 3.3 Which women return to their natal home, and do the characteristics of women returning to their natal home differ by state?

Older women are less likely to spend any part of the perinatal period in their natal home compared to younger women (Table 3). Compared to women under age 20, women aged 25–29 had about a 40% reduced odds of temporary childbirth migration across most timepoints, with some timepoints not meeting the criteria for statistical significance, and women 30+ had a 60%-70% reduced odds of temporary childbirth migration across timepoints. There was evidence of increased odds of migrating with increasing educational attainment, and this effect was stronger in Madhya Pradesh. Working outside of the home was associated with increased odds of returning to the natal home in Madhya Pradesh, but 30–40% reduced odds in Bihar across timepoints. Being Hindu is associated with reduced odds of returning to the natal home, especially in Madhya Pradesh. The index birth being the woman's first is associated with increased odds of returning to the natal home, especially in Bihar, but this is driven by being at the natal home during pregnancy, not for delivery or postpartum. There is some evidence that wealth was associated with increased odds of migrating, but results were not consistent across perinatal period and effect sizes were small. Odds of migrating was significantly patterned by district. S1 Table shows the odds of migration at different time points for both states combined.

## 3.4 Why do women return to their natal home?

*Quantitative findings*: About half of women who migrated (55%) say they returned to their natal home because they think they will receive better care, rest or treatment from their natal compared to husband's family (Table 4). Additionally, about 23% report that they thought that the quality of the services were better in their natal home. About 23% say that it was because their parents asked them to come. Fewer (16.7%) report that it is the culture/tradition (with more saying this in Madhya Pradesh), 11.2% because their husband's family asked them to go, 9.5% to avoid work in their husband's home (again more said this in Madhya Pradesh) and 4.3% because their husband and his family did not want to pay for services.

*Qualitative findings*: Mirroring the quantitative findings, women report usually going to their natal home in the last trimester of pregnancy, any time between months 7–9, and usually stayed there through the first 3 months after delivery. Most women describe the decision about whether or not she would return to her natal home as being made by their husband or in-laws. However, women often also mention a preference for returning to the natal home for delivery as they anticipated receiving better care (social support, attention) or being able to delay having the next birth for longer (especially if she had a daughter, which would lead to pressure to rapidly become pregnant again to bear a son). Women also discuss not having to do as much work at their natal home as in their marital home in the postpartum period. These perspectives are expressed in the quotations below:

> *In my maternal place, mother takes good care (of me), all necessary food items are there. (There is) no comparison of mother; mother in law is just mother in law, and she can't take place of real mother.*
>
> *(Age 19, 1 birth, currently pregnant, 10th grade education, Madhya Pradesh)*
>
> *Because there was no one to take care of me. . .. My husband is alone and he use to go to sell vegetables and I have my two year old daughter also, so my mother called me [asked me to come]*
>
> *(Age 26, 2 children, 12-24m postpartum, 5th grade education, Madhya Pradesh)*

## 3.5 What is the impact of returning to women's natal homes during the perinatal period on care access?

*Quantitative findings*: Among the full sample, 44% of women who returned to their natal home report that they saw either an AWW, ASHA or both AWW and ASHA while they were in their natal home. This is higher in Madhya Pradesh than in Bihar for every single time period, with a range of 53–64% of women reporting this in Madhya Pradesh and 18–35% in Bihar (Table 5).

*Antenatal Care*: Women in Bihar received a median of 2 ANC visits (Interquartile Range (IQR) 1–3) and women in Madhya Pradesh received significantly more visits, with a mean of 4 (IQR 2–6) (Table 6a). There was no significant difference in the median number of ANC visits by migration status, with non-migrating women having a mean of 3 (IQR 2–4) and migrating women a median of 3 (IQR 2–5) (Table 6b). In multivariable joint models incorporating the potential for selection into migration based on sociodemographic characteristics, women who spent more months in their natal home in the last trimester of pregnancy had a significantly higher ANC visit rate in Bihar but not in Madhya Pradesh (Table 7). Each month spent at the natal home in the third trimester of pregnancy was associated with a 5% and higher ANC rate in Bihar (Incidence Rate Ratio (IRR) 1.05, 95% CI 1.02–1.08). Women's education and household wealth were consistently positively associated with more ANC visits in both states, and ANC visits were patterned by district.

*Delivery*: A higher percentage of women who returned to their natal home (13.8%) delivered in a private facility, compared to those who did not (10.6%) (Table 6b). More women in Bihar delivered in a private facility compared to in Madhya Pradesh (11.1% compared to 5.1%) (Table 6a). However, in multivariable models accounting for selection into migration based on sociodemographic characteristics, women who were at their natal home at the time of delivery were no more likely to deliver in a private facility versus a public facility in both states (Table 7). Findings were consistent in supplementary analyses of home birth versus any facility

which found that women delivering in their natal homes were also no more likely to deliver in any facility (public or private) compared to home births (S2 Table). Education and wealth status were again consistently positively associated with both increased odds of delivering in a private vs. public facility and delivering at any health facility versus home. Again, private facility and any facility outcomes were patterned by district.

*Postnatal care*: About 50% of all women received 1 or more PNC visit, with no differences by migration status (Table 6b). Women in Madhya Pradesh were more likely to have one or more PNC visit compared to women in Bihar (48.7% vs. 25.8%) (Table 6a). Being in the natal home in the month immediately following delivery was not associated with PNC visits in either state in multivariable models accounting for potential sociodemographic selection effects (Table 7). Working outside of the home and wealth status were associated with higher odds of PNC visits, as was working outside the home, but in Bihar only. PNC visits were patterned by district.

*Qualitative Findings*: Anganwadi workers (AWWs) report that temporary childbirth migration is a substantial challenge for providing their care. They report that there is no way to track women who leave for their mother's village or identify women who come to their village because it was where their parents lived. A few AWWs also mentioned advancing the baby shower celebrations (part of the AWW services) for pregnant women before they leave for their natal homes, as a way of addressing childbirth migration. Most are unable to provide the appropriate counselling to women throughout the perinatal period, although some are able to work around this issue by through calling women on the phone or offering services earlier in pregnancy.

*"When I returned after four months (from my parents' home), the [AWW] came to know so she came after one month to make the card. . .. Yes. she could know only if I had informed her. Otherwise how would she know if I have returned or not."*

*(Beneficiary, age 21, 3rd birth, 6–12 month postpartum, 12th grade education, Bihar)*

*We do baby shower between the fifth month and the seventh months. Many times some [pregnant women] are left because she goes to her parent's place in the seventh month. . . In it (the baby shower) we (AWW) give all the things given in godh bharai (baby shower) like bangles, bracelets, nariyal (dry coconut) etc. That is why we do it in the fifth month and give all the information so that no one is left out.*

*(AWW, Age 45, Undergraduate education, Madhya Pradesh)*

AWWs also note that they are unable to enroll women temporarily at their natal homes in supplementary food programs provided by the government to pregnant women and newborns because they are not officially registered in their natal villages.

*How would I give them the take home ration (supplementary nutrition), already I have to choose from the many daughters-in-law in our village. This selection only creates a lot of problem, I can't even think of giving it to any daughter of the village (daughter of the village implies the women staying at their natal home). . ..*

*(AWW, Age 40 years, Undergraduate education, Bihar)*

## 4 Discussion

Temporary childbirth migration has the potential to disrupt the continuum of care. About one-third of women in our sample spent some part of their pregnancy, delivery, or postpartum

period away from their marital home where they are registered to receive antenatal and perinatal care. Most temporary childbirth migration was found to occur immediately before birth, at the time of delivery and in the early postpartum period, which are the most critical periods for the health of the mother and baby.

Despite our hypothesis that this disruption in the continuum of care would lead to fewer healthcare visits, we find the opposite—women who migrate to their natal home appear to receive more prenatal visits in one of the states of focus (Bihar), but there was no association with delivery in a private facility or any facility, or with postnatal care. Women in Madhya Pradesh did not have their care impacted by migrating. Our findings in Bihar may suggest that women's impetus for migrating may in fact play out; women believed that they would receive higher quality clinical care and more supportive care at their natal homes. In general, our results suggest that women who return to their natal home in pregnancy do get more care (or at least, more visits), perhaps due to higher autonomy/status in their natal home which prioritizes their care receipt irrespective of their ongoing access to free community-based services. Thus, women who migrate are likely to receive better or more essential services and information during pre or postnatal care and may be more likely to deliver in a facility. Past research has shown that women believe that private facilities provide better care, and thus they likely view delivery in a private facility as the preferred choice [46]. India has an incentive program for delivery in a public facility, and thus husband's families might pressure women to deliver at government facilities in order to receive the money. Some women noted that they went to the natal home because their husband's family did not want to pay for delivery (presumably in a private facility given delivery in public facilities is at no cost to the family).

These findings raise concern about the women left behind, at their marital home, and how to ensure that they receive appropriate, high quality, care. Older and higher parity women are less likely to return to their natal home, and these women may be at greater risk of health complications. It is possible that low women's status within their marital home, as has been found in previous studies, contributes to women being less able to access the services they need, despite being more connected to CHWs [60].

Our quantitative findings are in contrast to CHW narratives which described difficulties tracking women who temporarily migrate for childbirth. One of the main roles of CHWs is ensuring continuity of care across the perinatal period, and one goal of prenatal care is to get women into the "system" of that specific location, including registration in the local Anganwadi centre, a subcenter or a hospital, thereby increasing the likelihood of facility delivery. Thus, our finding that women who migrated generally received more care may reflect greater support and willingness of the natal family to ensure high quality care which overrides the barriers to care posed by lack of continuity of services. However, several questions remain about the impact of temporary childbirth migration on health outcomes that should be explored in future research on this topic, including mechanisms and impact on quality of care. For example, even if a woman can seek care in her natal home, if all of her previous visits were at another location, she might not have her complete medical record information and providers might not know about important risk factors, or even basic information about her health. While women registered with their local Anganwadi centers receive a Mother and Child Protection Card, women often do not carry them to their natal villages or might lose them.

Our findings that postnatal care (PNC) is not impacted by location similarly contradict the perspectives expressed by the AWWs, who expressed difficulties in identifying and providing care to women temporarily in their natal homes. It is possible that cultural practices of women staying in their homes for extended periods of time (mostly up to 40 days) in the postnatal

period contributes to this, making it harder for AWWs to access them or even know of their existence, regardless of whether they migrated or not. Also, PNC received by mothers within 48 hours after delivery is already low (57% and 42% in Madhya Pradesh and Bihar respectively) and additional PNC is infrequent in general, perhaps because it is not highly prioritized by the government system or families, and therefore any difference might be hard to detect [5]. Prospective designs and more nuanced quantitative and qualitative data are needed to disentangle these findings.

### 4.1 Explanations for state-level differences

One of the main findings of this analysis is that the pattern of maternal childbirth migration differs by states of India (even states that are relatively close to each other geographically), perhaps highlighting nuanced cultural practices during childbirth. Slightly more women returned to their natal home in Bihar compared to Madhya Pradesh; however, the difference was not statistically significant. While about 40% of women still reported some care from a CHW while at the natal home, the proportion was much smaller in Bihar compared to Madhya Pradesh. This suggests that not only does the prevalence of temporary childbirth migration differ by state, but what happens when women return to their natal home (in terms of connection to and continuity of care from their CHWs) differs. Gawde *et al.* (2016) in Mumbai, Maharashtra state, found that about two thirds of women returned home, however, this was a much smaller sample of women, all of who were migrants in an urban area [19]. Thus, collecting data from different regions among a diverse populations is an important next step for understanding this trend further, or within a specific population of interest.

Furthermore, we only found associations between being at the natal home and better health care use (for ANC) in Bihar, and not in Madhya Pradesh. Both Madhya Pradesh and Bihar fare poor on many maternal and child health indictors, as evidenced by both having a high burden of undernutrition indicated by high under-five mortality and anemia among pregnant and lactating women; however, compared to Madhya Pradesh, Bihar has worse indicators on health and nutrition program coverage and services including the Integrated Child Development Services (ICDS) [5]. These differing contexts of ICDS governance and implementation capabilities could help explain our findings. Bihar generally performs worse in most maternal, neonatal and child health indicators and services, and thus the added benefit of care and support (including emotional and financial) from the woman's natal home may make a bigger impact in Bihar [5]. On the other hand, in Madhya Pradesh, ICDS and maternal and neonatal health services are better, thus, women may be able to weather the disruption of the continuum of care period better. For example, 5.7% of women in Madhya Pradesh compared to 14.4% in Bihar received 4 ANC visits, 80% women in Madhya Pradesh deliver in facilities compared to 63.8% women in Bihar, and 17.5% of women in Madhya Pradesh compared to 10.8% in Bihar receive postnatal check-ups with 2 days [5]. Lower antenatal care and delivery uptake in Bihar is reflective of inadequate health infrastructure in Bihar, and therefore added support from natal families will make more of a difference. A recent Government of India's report documented systemic and structural deficiencies in Bihar's health system, including shortage of blood banks, shortage of beds, inadequate administration of tetanus toxoid injections to pregnant women, among other indicators [61]. Another explanation is that CHWs in Madhya Pradesh might more effectively connect with women who migrate, perhaps because it's a better functioning health care system broadly, making differences between the groups minimal in terms of coverage in Madhya Pradesh. Future research to understand the impacts of temporary childbirth migration should incorporate these multi-level factors.

## 4.2 Implications for research

Current data collection for household surveys in India including for the National Family Health Survey, do not account for temporary childbirth migration (specifically: if, where, when, duration or the impact of temporary childbirth migration is not collected). For household surveys collecting retrospective data on care received in the perinatal period and outcomes, this could lead us to misinterpret findings by geographic location. For example, a survey may report receipt of care by women in a district, but it does not mean that all these women received the care from CHWs of the same districts throughout her ANC and PNC.

Our analysis has shown that state level context mattered, with patterns and impacts differing between Bihar and Madhya Pradesh, so district or lower-level contexts should also matter. Indeed, our fixed-effects analyses identified some differences in migration behaviors and outcomes within district-specific comparisons which suggests that future research should further explore the role of district-specific contextual variables. About one-third of women had their natal home in a different district and one-third in a different block. Thus, for studies identifying women through registration lists from an Anganwadi center or a primary healthcare center, women not registered locally may be missed, and for studies where women are recruited at home, women not home when a data collector visits them where they usually live and are, presumably, not registered. Furthermore, specific factors may be associated with both temporary childbirth migration and adverse maternal and neonatal outcomes, resulting in biased findings without inclusion of these participants.

Temporary childbirth migration also causes problems for data collected from health facilities. If women are delivering at a different facility from where they received ANC, she might be recorded for the birth, and, given no previous health record at that facility, it might be assumed that she did not receive ANC, when in fact, she did, just at another location. While women do have a card that records their medical information, they may not always carry this or it may miss some information. This could be affecting our interpretation of the association between receiving care and delivery outcomes (complications, etc.). Finally, from census data, if a "household" is defined by people who have been sleeping there for a set amount of time, women may be left off, or double counted, or, potentially misclassified as being another type of migrant.

## 4.3 Strengths and limitations

This study has several strengths, including a large primary dataset collected from two Indian states which specifically asked about a variety of practices related to temporary childbirth migration, representing a more diverse sample of women and more comprehensive data collected than previous studies. Furthermore, we captured both quantitative and qualitative data from multiple stakeholders on this phenomenon. Our study also makes important contributions to the literature. Many studies that use Demographic Health Survey datasets focus on child health outcomes and relate them to exposure to pollution or shocks during month of birth but use residential location as the place where these shocks were observed (which may not be correct). However, data from this study suggest that this may not be an accurate assumption in all cases. It should be noted that a few studies have considered month of birth along with residential location [62,63].

However, several limitations do exist. First, we do not have a full sample of women who potentially migrate, as some women may have still been at their natal homes at the time of interview. We reduced the potential impact of this through limiting the sample to women who were at least 6 months postpartum; however, it is possible that certain women, who may be different systematically, were missing. Additionally, this restriction of the sample also reduced

the sample size. Second, we only include data from two states which are considered among severely under-resourced states in India, and our findings may not be generalizable outside of these states and populations. Additionally, our sample, which overrepresented rural and more disadvantaged populations, likely underestimates this trend overall since we find that more educated and richer women are more likely to migrate. Our data also come from women who are registered at Anganwadi Centres, so the results are only generalizable to women from these districts who are registered in the ICDS system. The NFHS-4 in India, which was collected around the time of this study, found that 65% of women who gave birth in the last 5 years in Bihar and 81% in Madhya Pradesh were registered [5]. Thus, these findings can only be interpreted as describing temporary childbirth migration and associated impacts on care among a population of women who are already in the health care system during their pregnancies. Women who are never registered in pregnancy may be unlikely to receive care at all, and thus might face no impacts of migrating during pregnancy (since they were never seeking care). On the other hand, women who do not register in their husband's home may migrate and somehow seek care in their natal home. More research is needed on this population. Data on health outcomes would add depth to our understanding of this phenomenon and should be collected by future studies. Given the hierarchical nature of our study sample and potential for endogeneity and selection bias, the models used for our primary analyses of the impact of temporary childbirth migration were complex, resulting in lower levels of precision around our effect estimates. As a result, our results may be conservative in identifying a true impact of migration on health outcomes. The qualitative study also purposively sampled women for understanding interaction with CHWs and did not specifically seek to explore the reasons and factors for childbirth migration. Finally, the retrospective nature of the data leads to the possibility of recall bias.

## 5 Conclusions

As a country, India still struggles with high rates of maternal and newborn mortality, and India contributes a high proportion of global maternal and newborn mortality deaths given its large population and high rates [64]. While India is diverse and heterogenous, if similar trends of temporary childbirth migration exist throughout the country, a substantial number of women are not connected to health services at some point during pregnancy. For example, of the estimated 24 million Indian births in 2019 [65], 32% returning to their natal home is equivalent to about 7.7 million women. Our findings suggest that the 16.3 million women giving birth in any given year who do <u>not</u> migrate are potentially less likely to access the care that they need and are in need of focused attention.

The phenomenon of temporary childbirth migration is potentially large, and requires much more attention in health care provision, analysis of factors related to the continuum of care and maternal and child health outcomes, and the demographic literature more broadly. Further research must be done to characterize it more fully. Temporary childbirth migration also likely has impacts for other south Asian countries as well, where we know anecdotally that it also occurs. Our findings have important implications for the Government of India's Maternal and Child Health Programs including the community health worker programs. Without a method for health workers to link women when they migrate, a substantial proportion of women may experience gaps in care during the perinatal period. Additionally, understanding why the women left behind appear to receive less care is critical, and likely intersects with women's household level empowerment in her marital home. Policies or programs may need to focus on reaching these women who are not able to migrate and have lower access to care from their natal home. From a data collection and demographic standpoint, many surveys that

we heavily rely on (such as the Demographic and Health Surveys) do not account for this phenomenon (or even ask about it), which could be responsible for misunderstandings about women's perinatal care experiences in these settings. New questions should be added about this into survey and routine data collection, including at health facilities. It is possible that better understanding this phenomenon will reveal previously missed opportunities to improve access to care for pregnant women and their newborns, thus improving maternal and child outcomes in India.

## Supporting information

**S1 Table. Odds of temporary childbirth migration by sociodemographic characteristics among women 6–12 months postpartum, Bihar and Madhya Pradesh combined, 2019.**
(DOCX)

**S2 Table. Relationship between temporary childbirth migration during the perinatal period and facility delivery among women 6–12 months postpartum, Bihar and Madhya Pradesh, 2019.**
(DOCX)

## Acknowledgments

We would like to thank the reviewers at the Bixby Center's Works in Progress at UCSF for their comments, Dr. Rasmi Avula, and Dr. Chuck McCulloch for his advice and assistance with our statistical analysis approach.

## Author Contributions

**Conceptualization:** Nadia Diamond-Smith, Sumeet Patil, Lia Fernald, Purnima Menon, Dilys Walker.

**Formal analysis:** Nadia Diamond-Smith, Alison M. El Ayadi.

**Funding acquisition:** Lia Fernald, Purnima Menon, Dilys Walker.

**Investigation:** Nadia Diamond-Smith, Lakshmi Gopalakrishnan, Sumeet Patil.

**Methodology:** Nadia Diamond-Smith, Sumeet Patil.

**Project administration:** Lakshmi Gopalakrishnan.

**Supervision:** Dilys Walker.

**Writing – original draft:** Nadia Diamond-Smith, Lakshmi Gopalakrishnan, Alison M. El Ayadi.

**Writing – review & editing:** Sumeet Patil, Lia Fernald, Purnima Menon, Dilys Walker.

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
