## [Decision Letter · Decision Letter 0]

5 Jul 2021

PONE-D-21-16583

Temporary childbirth migration and maternal health care in India

PLOS ONE

Dear Dr. Diamond-Smith,

Thank you for submitting your manuscript to PLOS ONE. I have now heard from two extremely knowledgeable reviewers who work in this area. Both recommended a revision. I have read the draft and given my interest in this topic (and region) and its policy relevance and I agree with their assessment.

I am grateful to the three reviewers who have provided excellent feedback and things you can work on (I hope you will appreciate their hard work). My take is comments from R1 and R2 are doable and it will make this paper very strong. I look forward to seeing the revised draft soon.

We look forward to receiving your revised manuscript.

Kind regards,

Nishith Prakash, Ph.D.

Academic Editor

PLOS ONE

Journal Requirements:

2. Thank you for stating in the manuscript methods section: 'Study protocols were reviewed and approved by institutional review boards at the University of California, Berkeley (Ref. No. 2016-08-9092), and the India-based Suraksha Independent Ethics Committee (Protocol No. 2016-08-9092). All participants provided informed consent. The trial is registered at https://doi.org/10.1186/ISRCTN83902145.'

Please provide additional details regarding participant consent. In the ethics statement in the Methods and online submission information, please ensure that you have specified what type you obtained (for instance, written or verbal, and if verbal, how it was documented and witnessed). If your study included minors, state whether you obtained consent from parents or guardians. If the need for consent was waived by the ethics committee, please include this information.

“This study was funded by Grant No. OPP1158231 from Bill and Melinda Gates Foundation (BMGF) to the University of California, San Francisco and University of California, Berkeley. The funders had no role in this analysis or manuscript. We would like to thank the reviews at the Bixby Center’s Works in Progress at UCSF for their comments, and Dr. Rasmi Avula.”

“This study was funded by Grant No. OPP1158231 from Bill and Melinda Gates Foundation (BMGF) to the University of California, San Francisco and University of California, Berkeley.”

Reviewers' comments:

Reviewer's Responses to Questions

**Comments to the Author**

1. Is the manuscript technically sound, and do the data support the conclusions?

Reviewer #1: Yes

Reviewer #2: Partly

2. Has the statistical analysis been performed appropriately and rigorously? 

Reviewer #1: Yes

Reviewer #2: Yes

3. Have the authors made all data underlying the findings in their manuscript fully available?

Reviewer #1: No

Reviewer #2: No

4. Is the manuscript presented in an intelligible fashion and written in standard English?

Reviewer #1: Yes

Reviewer #2: Yes

5. Review Comments to the Author

Reviewer #1: Overall, I think this paper is interesting and does paper does contribute to the literature by documenting an under investigated phenomena. The paper does contribute a documentation of an under investigated phenomena which potentially has some profound implications for policy. The analysis, although limited in scope does appear to be a reasonable approach. The discussion and conclusion do follow from the results. However, the paper does not extend much beyond documenting this phenomenon. Given that the main contribution of the paper is to document the phenomena, it may be worthwhile emphasising why this is important to document and what implications these results have, especially for health outcomes, which I believe are not covered sufficiently.

I can see that documenting this phenomenon is potentially important for policymakers and for future research. For researchers, not knowing that mothers migrate during pregnancy or soon thereafter is likely causing survey sampling issues because they migrate. Also, the authors make clear that many policies designed to support mothers during pregnancy are not designed to accommodate maternal migration, and therefore is important to document. Indeed, the authors say specialised healthcare workers say that there “is no way to track women who leave for their mother’s village or identify women who come to their village because it was where their parents lived.” In terms of future research, it is important to understand the health consequences on mothers and their children from migrating during pregnancy or postpartum. The analysis and discussion on health outcomes specifically is very limited, and I would have liked at least a discussion of likely impacts even if a formal analysis was not possible. Although, if associations between migration and health outcomes are possible to estimate they should be included. The discussion around health implications of this phenomena on mothers and babies is limited and I think this needs expanding, given that this is the key reason why it is important to document this phenomena.

I have several suggestions and comments which require attention prior to publication, none of which are particularly substantive but do require revisions to the manuscript. The comments are presented chronologically.

Section 1: Is there any evidence of childbirth migration in developed countries? If so, do motivations differ? This may not be relevant but add if there is literature on this.

Line 147 to line 152: I think these beliefs of ritual pollution warrants a little more discussion. Could it be that this is a driver of migration? If mother “should only be touched by close female relatives including grandmothers” then this could almost fully explain reasons for returning to natal home. It would be useful to provide a statistic of how widespread of a belief this is, and a very small amount more explanation of the belief and its repercussions. It is interesting that mothers do not mention it as a motivator for moving. I would have also expected this to be included specifically in the quantitative component of the analysis, not simply “for cultural reason”. It could be that this type of belief is driving mothers to their natal homes because they receive no care in their family home as a result of these beliefs. If these beliefs mean they receive no care in their family home, then it makes sense that mothers would report that they “receive better care” in their natal home, but it really is the belief system which is driving these results. Interviewees may not even considered this a driver because the belief system is so fundamental to them, and therefore the “better care” reason for them migrating is actually driven by these beliefs. I think this point is not sufficiently addressed. Even the authors believe that these beliefs are not driving the phenomena, I’d still like to be presented a reason why and I think it would be useful to have at least a brief discussion on this considering the importance. Was this mentioned at all in the interviews? If so, I would like it to be included.

Line 170: “women only represent 1.1% of women’s migration”

This sentence doesn’t make sense to me.

Line 203: “Two cadres of all-female CHWs work at the community-level delivering last mile services to pregnant, lactating women and infants – the Accredited Social Health Activist (ASHA) under National Health Mission and the Anganwadi Worker (AWW) under India’s flagship nutrition program, the Integrated Child Development Services [44,45].”

This sentence could be more clearly written.

Line 217: “we estimate at the impact of being in the natal home,”

I don’t think the ‘at’ is needed here.

Section 2.1: It would be useful to know how Madhya Pradesh and Bihar are different to one another, ideally using general descriptive statistics of the state, and how representative these states are of India as a whole. It would be useful to know a priori how we expect the results from these two states to differ.

Section 2.1: The authors should include reasons why these states were chosen for this analysis, because it is not clear as to why these specific states were chosen above all others.

Line 234: “then up to two AWCs/AWWs were sampled per village Then we randomly sampled up to eight mothers”

I think a full stop is needed here or change of character case.

Section 2.1.1: Is there no other source of selective sampling apart from apart from higher probability of sampling non-migrating mothers for your approach? Do some mothers “slip through the net” of the AWC/AWW system? If some mothers do slip through the net, then who are these likely to be and how would that impact your results and findings? The entire sample comes from those that engage with the AWC/AWW system. I think this is a major source of sample selection, and is a point not sufficiently addressed or consider.

Line 238,239: “interpreted this as indicating that by 6 months postpartum the vast majority of women have returned to their marital home.”

Interpreted what exactly? Authors should be clearer. It’s not clear on what you’re interpreting or how you reach this conclusion.

Line 271: “We do this because the vast majority of women had met the Indian criteria of 4 or more, we wanted to highlight those that exceeded that threshold.”

Are the authors able to show descriptive statistics of ANC visits, specifically what is meant by “vast majority of women”? It would be useful to see a granular distribution of this i.e. what % received 1 visit, what % received 2 visits, % that received 3 visits etc. because this seems like a peculiar discretisation decision to make, given that the guidelines are different to the authors categorisation. Your current analysis is “more than sufficient” vs. “sufficient or less”, which is a strange discretisation in my opinion. “insufficient” vs. “sufficient” ANC visits would be more of an obvious discretisation, and your deviation from this is not well supported. It is a little strange that the reason for the discretisation is not supported with empirical evidence in the paper i.e. distribution of visits, so that the reader can see for themselves what “vast majority” means, and this would make the discretisation more convincing. Simply presenting whether received “Received 5 or more ANC visits” is not a sufficient descriptive statistic to justify this discretisation decision.

Section 2.1.3: Are all the independent variables estimates jointly in the same regression? I presume so, but it does not say explicitly.

Line 365: “Working outside of the home is associated with increased odds of returning to the natal home in Madhya Pradesh but reduces the odds of returning home in Bihar.”

This is an example of a results which is perplexing, and if descriptive statistics and further background was presented for each state, then reader would be more prepared to understand these results. This happens a few times in the paper, and it’s difficult to understand given that there is limited background on each state.

Line 410: “About 30% of women who returned home and those did not return home received 5 or more ANC visits (Table 5)”

This sentence could be written more clearly, because it doesn’t make much sense to me.

Line 414: “Similar analyses for fewer ANC visits showed no effect (not shown).”

I think this is simply the inverse of “receiving 5 or more ANC visits”. How would these results be different to the ones from “receiving 5 or more ANC visits”, apart from the change in sign. It is not clear what this adds to your results.

Line 424,425: “multinomial models show no difference between private and public or home (also not shown).”

Even if they are not statistically significant, I would prefer if these results were not supressed and instead reported in the paper alongside the other results.

Line 435: Are the qualitative results in contradiction to the empirical ones? AWWs say that they can’t track mothers, and they find it harder to provide their services but being at natal address increasing odds of receiving an ANC visit. I realise these services are distinct, but if migrating mothers fall through the healthcare system, then how can it be that ANC visits are more likely to happen. It would be useful to have some discussion around this point, or to provide a little more background on the difference between AWWs and ANC services, because that might make these results more intuitive. These findings make me a little confused, and it may be insufficient background. This point is partially addressed in the discussion, but the authors don’t provide any reason why this might be the case, and I think this is important because I conclude that it is some form of statistical bias in your estimates without a reasonable explanation.

Section 5, line 465: I think the number of this heading is incorrect.

Section 5: It would be useful to discuss the overall health consequences of childbirth migration. Presumably the migration itself could impact health of mother and baby negatively, due to the stresses involved with migration itself. On the other hand, better care is provided at the natal home. Authors should discuss the overall health consequences of temporary childbirth migration.

Section 5: The authors should discuss policy implications. Specifically, it would be useful to know what these results mean for prenatal and postpartum healthcare provision, and how these issues could be tackled.

Section 5.1: It would be very useful to have some background on the differences between these states when describing the data earlier, so that the readers know what to expect when it comes to these differences. I have made this point already but, it is not clear how these states differ substantively prior to this discussion, and how these states differ to the average Indian state.

Reviewer #2: Temporary Childbirth Migration and Maternal Healthcare in India

Overview: The paper addresses a very important and relevant research question related to migration of women due to childbirth in two states of India – Madhya Pradesh & Bihar). This phenomenon has been largely understudied and undocumented. The paper provides analysis (both quantitative & qualitative) related to three major things:

a) Prevalence of this phenomenon

b) Socio-economic determinants for this phenomenon

c) Quantity of healthcare received by mothers due to this phenomenon

While the paper describes the prevalence of this phenomenon quite clearly, I have concerns about the way sample was selected for subsequent analysis (see point 1 under major comments). The analysis will benefit from robustness checks described below which can make the results more believable. Lastly, the results are not consistent across states (for both determinants of migration (in Table 2) and healthcare received (in Table 6)), this apparent difference in results has been discussed qualitatively (in section 5.1) but have not been quantitatively supported.

Detailed Comments:

Major:

1) The results related to socio-economic determinants of temporary child birth migration (Tables 2) uses a sample which chooses women who are 6 months post-delivery (child age > 6m) however this sample selection leads to a substantial drop in observations. A robustness check designed around this cut-off (time elapsed period post-delivery) for sample selection will help in establishing that the results are not sensitive to a particular cut-off that was chosen for analysis. Based on discussion in section (lines 384 to 386; 355) it makes sense to choose a different cut-off which is 3 months or 1 month post-delivery.

2) The empirical analysis introduces many control variables however an important determinant in form of district fixed effects is missing. Since majority of women might be migrating within the same district (lines 343 to 346) hence it makes sense to use district fixed effects which accounts for time-invariant heterogeneity (health infrastructure in a district etc). Most of the major health programs focused on child and maternal health are implemented in a top-down fashion where districts submit their expected budgets to the health ministry, the funds are also allocated to a district. To a great extent implementation of health policies in India are run/directed by the district administration. So, a strong suggestion is to include these fixed effects in all analyses that have been carried out in this paper.

3) How many women (in each state) out of the total sampled women participated in temporary childbirth migration? The analysis can be complemented with a selection model wherein in the first stage decision to migrate is modelled and in the second stage healthcare outcomes are modelled.

4) The reasons provided for difference in results for the two states (results are present for MP but not for Bihar) remain unclear. These reasons should be elaborated further and if possible, mechanisms should be explored which explain the difference in results for the two states. Perhaps NFHS dataset (which is representative at district level) can be used to document this systematic difference in healthcare delivery in these two states (specifically in the 12 districts which are the focus of this study).

5) Table 6: Explores the relationship “Five or more ANC visits” and “Being at natal home during last month of pregnancy”. Shouldn’t the dependent variable be number of ANC visits during last month of pregnancy. Most of the ANC visits in this scenario (where a mother only migrated during last month of the pregnancy) would have already taken place in her marital home rather than natal home.

Minor:

Overall, the writing is lucid but, in some places, it will benefit from further elaboration/clarification.

- Line 526: “Slightly more women returned to their natal home in Bihar compared to Madhya Pradesh, however the difference was of practically meaningful” Does this mean that the difference between the states was statistically insignificant? If yes, then it contradicts the statement in line 538: “First, we see that more women return to their natal home in Bihar”.

- Lines 607 to 610 provide a back of the envelope calculation for number of women who face disrupted perinatal care due to temporary childbirth migration. However, this is confusing and contradicts the main findings of the paper (for Madhya Pradesh) which show that despite temporary childbirth migration, women received more care. Even for Bihar the null result signifies no drop in the level of care. Hence, in light of the results of the paper (and conclusion as mentioned in abstract) can the authors claim that there has been a disruption in health care for mothers who participated in temporary childbirth migration?

- Add another very important contribution to the literature: many papers which use Demographic Health Survey (DHS) datasets focus on child health outcomes and relate them to exposure to pollution or shocks during month of birth but use residential location as the place where these shocks were observed (which may not be correct). In light of this study, an important factor is revealed to the researchers in form of substantial proportion of mothers migrating to their native place during month of birth of a child. Few studies which have used month of birth along with residential location include – Spears, Menon etc.

*Brainerd, E., & Menon, N. (2014). Seasonal effects of water quality: The hidden costs of the Green Revolution to infant and child health in India. Journal of Development Economics, 107, 49-64.

* Spears, D., Dey, S., Chowdhury, S., Scovronick, N., Vyas, S., & Apte, J. (2019). The association of early-life exposure to ambient PM 2.5 and later-childhood height-for-age in India: an observational study. Environmental Health, 18(1), 1-10.

- Line 351 – The figure about one-third women (32.4%) participating in temporary childbirth migration cannot be located in Figure 2 (as referenced in the manuscript).

- I am not very clear about propensity score matching mentioned in line 233 in section 2.1.1. Why was it used? What was the dependent variable?

- (This is optional) Can the authors include actual empirical model in the manuscript i.e. write the model equations which are used for analysis?

- Discrepancy in line 242? Number of observations mentioned as n = 3055 but this figure doesn’t match with 782+1273.

- Tables 3, 4: Are the significance levels indicative of difference between states? Mention this detail in table footnotes.

- Table 5: Different significance definition mentioned in table footnotes. Please use uniform significance level definitions throughout the paper.

- Line 422: Should be “….more likely…” rather than “more like”.

6. PLOS authors have the option to publish the peer review history of their article (what does this mean?). If published, this will include your full peer review and any attached files.

Reviewer #1: No

Reviewer #2: **Yes: **Prachi Singh

---

## [Author Response · Author response to Decision Letter 0]

14 Oct 2021

Comments to the Author

1. Is the manuscript technically sound, and do the data support the conclusions?

Reviewer #1: Yes

Reviewer #2: Partly

2. Has the statistical analysis been performed appropriately and rigorously?

Reviewer #1: Yes

Reviewer #2: Yes

3. Have the authors made all data underlying the findings in their manuscript fully available?

Reviewer #1: No

Reviewer #2: No

4. Is the manuscript presented in an intelligible fashion and written in standard English?

Reviewer #1: Yes

Reviewer #2: Yes

5. Review Comments to the Author

Reviewer #1: Overall, I think this paper is interesting and does paper does contribute to the literature by documenting an under investigated phenomena. The paper does contribute a documentation of an under investigated phenomena which potentially has some profound implications for policy. The analysis, although limited in scope does appear to be a reasonable approach. The discussion and conclusion do follow from the results. However, the paper does not extend much beyond documenting this phenomenon. Given that the main contribution of the paper is to document the phenomena, it may be worthwhile emphasising why this is important to document and what implications these results have, especially for health outcomes, which I believe are not covered sufficiently.

I can see that documenting this phenomenon is potentially important for policymakers and for future research. For researchers, not knowing that mothers migrate during pregnancy or soon thereafter is likely causing survey sampling issues because they migrate. Also, the authors make clear that many policies designed to support mothers during pregnancy are not designed to accommodate maternal migration, and therefore is important to document. Indeed, the authors say specialised healthcare workers say that there “is no way to track women who leave for their mother’s village or identify women who come to their village because it was where their parents lived.” In terms of future research, it is important to understand the health consequences on mothers and their children from migrating during pregnancy or postpartum. The analysis and discussion on health outcomes specifically is very limited, and I would have liked at least a discussion of likely impacts even if a formal analysis was not possible. Although, if associations between migration and health outcomes are possible to estimate they should be included. The discussion around health implications of this phenomena on mothers and babies is limited and I think this needs expanding, given that this is the key reason why it is important to document this phenomena.

I have several suggestions and comments which require attention prior to publication, none of which are particularly substantive but do require revisions to the manuscript. The comments are presented chronologically.

Section 1: Is there any evidence of childbirth migration in developed countries? If so, do motivations differ? This may not be relevant but add if there is literature on this.

We searched the literature but could not find any childbirth migration related publications from developed countries.

Line 147 to line 152: I think these beliefs of ritual pollution warrants a little more discussion. Could it be that this is a driver of migration? If mother “should only be touched by close female relatives including grandmothers” then this could almost fully explain reasons for returning to natal home. It would be useful to provide a statistic of how widespread of a belief this is, and a very small amount more explanation of the belief and its repercussions. It is interesting that mothers do not mention it as a motivator for moving. I would have also expected this to be included specifically in the quantitative component of the analysis, not simply “for cultural reason”. It could be that this type of belief is driving mothers to their natal homes because they receive no care in their family home as a result of these beliefs. If these beliefs mean they receive no care in their family home, then it makes sense that mothers would report that they “receive better care” in their natal home, but it really is the belief system which is driving these results. Interviewees may not even considered this a driver because the belief system is so fundamental to them, and therefore the “better care” reason for them migrating is actually driven by these beliefs. I think this point is not sufficiently addressed. Even the authors believe that these beliefs are not driving the phenomena, I’d still like to be presented a reason why and I think it would be useful to have at least a brief discussion on this considering the importance. Was this mentioned at all in the interviews? If so, I would like it to be included.

Thank you for this comment. We have addressed it by adding more literature surrounding ritual pollution. While there is no statistic available to quantify the phenomenon, it has been documented in other Asian settings such as Nepal, Pakistan, etc. We added those studies to expand on the topic. 

We also would like to note that we don’t have ritual pollution as one of the answers in our structured surveys. 

Line 170: “women only represent 1.1% of women’s migration”

This sentence doesn’t make sense to me.

Addressed

Line 203: “Two cadres of all-female CHWs work at the community-level delivering last mile services to pregnant, lactating women and infants – the Accredited Social Health Activist (ASHA) under National Health Mission and the Anganwadi Worker (AWW) under India’s flagship nutrition program, the Integrated Child Development Services [44,45].”

This sentence could be more clearly written.

Addressed

Line 217: “we estimate at the impact of being in the natal home,”

I don’t think the ‘at’ is needed here.

Change made

Section 2.1: It would be useful to know how Madhya Pradesh and Bihar are different to one another, ideally using general descriptive statistics of the state, and how representative these states are of India as a whole. It would be useful to know a priori how we expect the results from these two states to differ.

We added a note on the differences between MP and Bihar based on other data such as DHS. 

Section 2.1: The authors should include reasons why these states were chosen for this analysis, because it is not clear as to why these specific states were chosen above all others.

We have addressed this in the methods section.

Line 234: “then up to two AWCs/AWWs were sampled per village Then we randomly sampled up to eight mothers”

I think a full stop is needed here or change of character case.

We have addressed this oversight.

Section 2.1.1: Is there no other source of selective sampling apart from apart from higher probability of sampling non-migrating mothers for your approach? Do some mothers “slip through the net” of the AWC/AWW system? If some mothers do slip through the net, then who are these likely to be and how would that impact your results and findings? The entire sample comes from those that engage with the AWC/AWW system. I think this is a major source of sample selection, and is a point not sufficiently addressed or consider.

Line 238,239: “interpreted this as indicating that by 6 months postpartum the vast majority of women have returned to their marital home.”

Interpreted what exactly? Authors should be clearer. It’s not clear on what you’re interpreting or how you reach this conclusion.

Thank you for pointing this out. I think that this sentence somehow got misplaced and we have now moved it to the end of that paragraph, since it was referring to our interpretation of Fig 2. 

Line 271: “We do this because the vast majority of women had met the Indian criteria of 4 or more, we wanted to highlight those that exceeded that threshold.”

Are the authors able to show descriptive statistics of ANC visits, specifically what is meant by “vast majority of women”? It would be useful to see a granular distribution of this i.e. what % received 1 visit, what % received 2 visits, % that received 3 visits etc. because this seems like a peculiar discretisation decision to make, given that the guidelines are different to the authors categorisation. Your current analysis is “more than sufficient” vs. “sufficient or less”, which is a strange discretisation in my opinion. “insufficient” vs. “sufficient” ANC visits would be more of an obvious discretisation, and your deviation from this is not well supported. It is a little strange that the reason for the discretisation is not supported with empirical evidence in the paper i.e. distribution of visits, so that the reader can see for themselves what “vast majority” means, and this would make the discretisation more convincing. Simply presenting whether received “Received 5 or more ANC visits” is not a sufficient descriptive statistic to justify this discretisation decision.

Thank you for this comment. We have now added a graph that shows the distribution of ANC visits. As you can see, most women get 2, 3 or 4 visits, with 42% of women meeting the guidelines of getting 4 or more ANC. We wanted to reflect the fact that WHO guidelines recommend more ANC visits (8) than the Indian government (4), and that most of these should occur in the last months of pregnancy. Thus, we felt that TCM will likely have the largest impact on those later ANC visits, since women migrate in the later stages of pregnancy. We have added the following in the document: ”For pregnancy-related analyses, we operationalized natal home during pregnancy as being at the natal home in the last month of pregnancy to capture the full extent of possible antenatal care. The Indian government officially recommends a minimum of 4 ANC visits, while the WHO recently recommended 8 ANC visits (Centre for Health Informatics (CHI), 2016). At least of these visits is supposed to be in the last month of pregnancy. Most of these higher order (4+) visits will occur in the later stages of pregnancy, when women are more likely to be in their natal village. We operationalized achievement of ANC as 5 or more ANC visits compared to 4 or fewer. As can be seen in Figure 2 most women do not meet the Indian guidelines, and only 25% receive more than 4 ANC visits. Since global guidelines recommend more visits, and we know that these visits are most likely to occur at the end of pregnancy, our indicator was constructed to measure women obtaining ANC visits above the national recommended guidelines and in the last month of pregnancy.” 

Section 2.1.3: Are all the independent variables estimates jointly in the same regression? I presume so, but it does not say explicitly.

Yes, we have clarified this now by adding “…all incorporating sociodemographic covariates, estimated jointly.”

Line 365: “Working outside of the home is associated with increased odds of returning to the natal home in Madhya Pradesh but reduces the odds of returning home in Bihar.”

This is an example of a results which is perplexing, and if descriptive statistics and further background was presented for each state, then reader would be more prepared to understand these results. This happens a few times in the paper, and it’s difficult to understand given that there is limited background on each state.

Indicators on ANC, immunization, nutrition, female literacy, etc. has been added to give the readers state-specific context. 

Line 410: “About 30% of women who returned home and those did not return home received 5 or more ANC visits (Table 5)”

This sentence could be written more clearly, because it doesn’t make much sense to me.

We have edited this to say “About 30% of all women...”

Line 414: “Similar analyses for fewer ANC visits showed no effect (not shown).”

I think this is simply the inverse of “receiving 5 or more ANC visits”. How would these results be different to the ones from “receiving 5 or more ANC visits”, apart from the change in sign. It is not clear what this adds to your results.

Apologies, the wording was confusing, we meant to say that we did an analysis looking at 4 or more and 3 or more ANC visits and the results were the same. We have re-worded it to read “Similar analyses for receiving fewer ANC visits (4 or more, for example) showed..”

Line 424,425: “multinomial models show no difference between private and public or home (also not shown).”

Even if they are not statistically significant, I would prefer if these results were not supressed and instead reported in the paper alongside the other results.

Thank you for this comment. We did consider adding these findings, but ultimately, we worried that we are already presenting a lot and perhaps too much in this paper, and felt that this would complicate things further. However, we did want to show that we were rigorous in our analysis and explored other outcomes, but presented those that were most grounded in our theory of what migration might impact. We would strongly prefer to leae this as is, but can remove the reference to the additional analyses if the editors feel that we should. 

Line 435: Are the qualitative results in contradiction to the empirical ones? AWWs say that they can’t track mothers, and they find it harder to provide their services but being at natal address increasing odds of receiving an ANC visit. I realise these services are distinct, but if migrating mothers fall through the healthcare system, then how can it be that ANC visits are more likely to happen. It would be useful to have some discussion around this point, or to provide a little more background on the difference between AWWs and ANC services, because that might make these results more intuitive. These findings make me a little confused, and it may be insufficient background. This point is partially addressed in the discussion, but the authors don’t provide any reason why this might be the case, and I think this is important because I conclude that it is some form of statistical bias in your estimates without a reasonable explanation.

We have sought to clarify this seeming discrepancy through adding further material throughout the manuscript. First, we have expanded our introduction on the roles of community-based health workers in the Indian context, and added text explaining catchment populations and eligibility for services, and how CHW awareness of populations in need of care who are temporarily in her catchment area may be low. Then in the discussion we have added further text suggestion that the natal household may prioritize perinatal care receipt despite the challenges around ongoing access to CHWs because of women’s higher autonomy within this environment. We hope that this additional explanation provides enough clarity to the reviewer. 

Section 5, line 465: I think the number of this heading is incorrect.

Thank you for noting this, they were incorrect in many places! We have edited. 

Section 5: It would be useful to discuss the overall health consequences of childbirth migration. Presumably the migration itself could impact health of mother and baby negatively, due to the stresses involved with migration itself. On the other hand, better care is provided at the natal home. Authors should discuss the overall health consequences of temporary childbirth migration.

Given the limitations of our data, we are unable to extend our findings beyond health service use to actual health outcomes; however, we agree with the reviewer that this is an important next step for understanding the phenomenon and its implications, and are planning to continue our work in this area to develop a robust understanding of the health outcome impact. 

Section 5: The authors should discuss policy implications. Specifically, it would be useful to know what these results mean for prenatal and postpartum healthcare provision, and how these issues could be tackled.

We have added to the discussion section where we discuss this, which now reads “Our findings have important implications for the Government of India’s Maternal and Child Health Programs including the community health worker programs. Without a method for health workers to link women when they migrate, a substantial proportion of women may experience gaps in care during the perinatal period. Additionally, understanding why the women left behind appear to receive less care is critical, and likely intersects with women’s household level empowerment in her marital home. Policies or programs may need to focus on reaching these women who are not able to migrate and have high quality care from their natal home. From a data collection and demographic standpoint, many surveys that we heavily rely on (such as the Demographic and Health Surveys) do not account for this phenomenon (or even ask about it), which could be responsible for misunderstandings about women’s perinatal care experiences in these settings. New questions should be added about this into survey and routine data collection, including at health facilities.”

Section 5.1: It would be very useful to have some background on the differences between these states when describing the data earlier, so that the readers know what to expect when it comes to these differences. I have made this point already but, it is not clear how these states differ substantively prior to this discussion, and how these states differ to the average Indian state.

We have added this from nationally representative survey (DHS 2015-16).

Reviewer #2: Temporary Childbirth Migration and Maternal Healthcare in India

Overview: The paper addresses a very important and relevant research question related to migration of women due to childbirth in two states of India – Madhya Pradesh & Bihar). This phenomenon has been largely understudied and undocumented. The paper provides analysis (both quantitative & qualitative) related to three major things:

a) Prevalence of this phenomenon

b) Socio-economic determinants for this phenomenon

c) Quantity of healthcare received by mothers due to this phenomenon

While the paper describes the prevalence of this phenomenon quite clearly, I have concerns about the way sample was selected for subsequent analysis (see point 1 under major comments). The analysis will benefit from robustness checks described below which can make the results more believable. Lastly, the results are not consistent across states (for both determinants of migration (in Table 2) and healthcare received (in Table 6)), this apparent difference in results has been discussed qualitatively (in section 5.1) but have not been quantitatively supported.

Detailed Comments:

Major:

1) The results related to socio-economic determinants of temporary child birth migration (Tables 2) uses a sample which chooses women who are 6 months post-delivery (child age > 6m) however this sample selection leads to a substantial drop in observations. A robustness check designed around this cut-off (time elapsed period post-delivery) for sample selection will help in establishing that the results are not sensitive to a particular cut-off that was chosen for analysis. Based on discussion in section (lines 384 to 386; 355) it makes sense to choose a different cut-off which is 3 months or 1 month post-delivery.

Thank you for this suggestion. We felt that given Figure 2, the most substantial drop was after 6 months and thus did not want to limit the sample to women in a narrower window. We have added the following “We ran a sensitivity analysis (robustness check) on all models using the full sample and there were no significant differences.”

2) The empirical analysis introduces many control variables however an important determinant in form of district fixed effects is missing. Since majority of women might be migrating within the same district (lines 343 to 346) hence it makes sense to use district fixed effects which accounts for time-invariant heterogeneity (health infrastructure in a district etc). Most of the major health programs focused on child and maternal health are implemented in a top-down fashion where districts submit their expected budgets to the health ministry, the funds are also allocated to a district. To a great extent implementation of health policies in India are run/directed by the district administration. So, a strong suggestion is to include these fixed effects in all analyses that have been carried out in this paper.

All analyses are clustered at the village level, which should account for much of these concerns. We ran a sensitivity analysis with district included as a fixed effect and it did not alter the findings of the presented analysis. We have noted the following :” Sensitivity analyses were run to include “district” as a fixed effect, and results were not substantially different.”

3) How many women (in each state) out of the total sampled women participated in temporary childbirth migration? The analysis can be complemented with a selection model wherein in the first stage decision to migrate is modelled and in the second stage healthcare outcomes are modelled.

A total of 2,921 women overall participated in any temporary childbirth migration, 1,753 in Bihar and 1,168 in Madhya Pradesh. We have added these numbers to the text. 

We appreciate the reviewer’s suggestion for extending our analysis, and have developed joint models incorporating selection into temporary childbirth selection for each of our outcomes of interest using a generalized structural equation modeling approach. We ran these models and there were no changes in the relationships between temporary childbirth migration and the outcomes, and thus we have not changed our models in the paper. We have however added that we ran this as a sensitivity analysis in the paper itself “We also ran joint models incorporating selection into temporary childbirth selection for each of our outcomes of interest using a generalized structural equation modeling approach as another sensitivity analysis. These models did not differ substantially from the models presented and thus are not shown.”

4) The reasons provided for difference in results for the two states (results are present for MP but not for Bihar) remain unclear. These reasons should be elaborated further and if possible, mechanisms should be explored which explain the difference in results for the two states. Perhaps NFHS dataset (which is representative at district level) can be used to document this systematic difference in healthcare delivery in these two states (specifically in the 12 districts which are the focus of this study).

We have added a little more detail into the section about state-level differences in the discussion section, including the following specifically about health system/delivery factors “Health system factors could factor into the differences we see by state, for example, 35.7% of women in MP compared to 14.4% in Bihar received 4 ANC visits, and 17.5% of women in MP compared to 10.8% in Bihar receive postnatal check-ups with 2 days(International Institute of Population Sciences, 2017). “

5) Table 6: Explores the relationship “Five or more ANC visits” and “Being at natal home during last month of pregnancy”. Shouldn’t the dependent variable be number of ANC visits during last month of pregnancy. Most of the ANC visits in this scenario (where a mother only migrated during last month of the pregnancy) would have already taken place in her marital home rather than natal home.

Yes, we agree that some of these visits may have occurred before the woman migrated, however, what we know about care seeking in pregnancy in India (and elsewhere) is that few women have visits early in pregnancy, and the number of visits increases in later pregnancy. A different (and interesting) analysis could be to see if women who migrated early in pregnancy were more likely to have a visit in the first trimester (which is an important gap), however, we did not do so in this analysis because of sample size issues with the number of women migrating in early pregnancy and because we were interested in using a standard measure of appropriate/adequate ANC visits. We had the exposure be “being at the natal home in the last month of pregnancy” to make sure that we were comparing appropriate groups of women to each other. For example, if we looked at women who were at the natal home in month 6, they might not have had enough time to get the adequate number of visits. By looking at women in the last month of pregnancy we are able to ensure that all women had comparable time to have enough visits. 

Minor:

Overall, the writing is lucid but, in some places, it will benefit from further elaboration/clarification.

- Line 526: “Slightly more women returned to their natal home in Bihar compared to Madhya Pradesh, however the difference was of practically meaningful” Does this mean that the difference between the states was statistically insignificant? If yes, then it contradicts the statement in line 538: “First, we see that more women return to their natal home in Bihar”.

We have edited this for clarity to read “statistically significant”

- Lines 607 to 610 provide a back of the envelope calculation for number of women who face disrupted perinatal care due to temporary childbirth migration. However, this is confusing and contradicts the main findings of the paper (for Madhya Pradesh) which show that despite temporary childbirth migration, women received more care. Even for Bihar the null result signifies no drop in the level of care. Hence, in light of the results of the paper (and conclusion as mentioned in abstract) can the authors claim that there has been a disruption in health care for mothers who participated in temporary childbirth migration?

We have revised this to read as follows “Our findings suggest that the 16.3 million women giving birth in any given year who do not migrate are potentially less likely be access the care that they need and are in need of focused attention.”

- Add another very important contribution to the literature: many papers which use Demographic Health Survey (DHS) datasets focus on child health outcomes and relate them to exposure to pollution or shocks during month of birth but use residential location as the place where these shocks were observed (which may not be correct). In light of this study, an important factor is revealed to the researchers in form of substantial proportion of mothers migrating to their native place during month of birth of a child. Few studies which have used month of birth along with residential location include – Spears, Menon etc.

*Brainerd, E., & Menon, N. (2014). Seasonal effects of water quality: The hidden costs of the Green Revolution to infant and child health in India. Journal of Development Economics, 107, 49-64.

* Spears, D., Dey, S., Chowdhury, S., Scovronick, N., Vyas, S., & Apte, J. (2019). The association of early-life exposure to ambient PM 2.5 and later-childhood height-for-age in India: an observational study. Environmental Health, 18(1), 1-10.

Added this under strengths of the study. 

- Line 351 – The figure about one-third women (32.4%) participating in temporary childbirth migration cannot be located in Figure 2 (as referenced in the manuscript).

Apologies, the reference to Figure 2 should be after the next sentence, we have moved it. 

- I am not very clear about propensity score matching mentioned in line 233 in section 2.1.1. Why was it used? What was the dependent variable?

We have added more details to explain the variables used in PSM, in a foot note, see below. We have added a reference to the study protocol that explains this is more depth. 

“Census 2011 variables used for matching: Distance between village and block headquarters (kilometers); Population; % of SC/ST households; % of villages served by public transport; % of villages connected to a major road; % of villages with a public ration shop; % of villages with a post office; % of villages with a bank; Average proportion of households in a village with a bank account; % of villages with an agricultural society; % of villages with a self-help group; Average proportion of households in a village serviced by closed drainage system; Average proportion of households in a village with improved source of drinking water; Average proportion of households in a village with improved sanitation facility; Average proportion of households in a village using electricity as the main source of light; Average proportion of households in a village with a pucca house; Average household asset index for the village”

- (This is optional) Can the authors include actual empirical model in the manuscript i.e. write the model equations which are used for analysis?

We have added the empirical models which were used for analysis into the methods section. 

- Discrepancy in line 242? Number of observations mentioned as n = 3055 but this figure doesn’t match with 782+1273.

We have edited this, thank you! (n=3,121; 1,832 in Bihar and 1,289 in Madhya Pradesh)

- Tables 3, 4: Are the significance levels indicative of difference between states? Mention this detail in table footnotes.

No, in those tables the significance is indicative of factors associated with the outcomes, as listed in the header of each column. 

- Table 5: Different significance definition mentioned in table footnotes. Please use uniform significance level definitions throughout the paper.

Edited

- Line 422: Should be “….more likely…” rather than “more like”.

Edited

---

## [Decision Letter · Decision Letter 1]

7 Dec 2021

PONE-D-21-16583R1Temporary childbirth migration and maternal health care in IndiaPLOS ONE

Dear Nadia , Thank you for sending the revised version. This definitely looks better, however, both referees were not convinced completely. They have made tons of suggestions which you should take seriously. My own read of the paper and the responses were that it was not very transparent and clear. I hope you will address all comments during this revision.

We look forward to receiving your revised manuscript.

Kind regards,

Nishith Prakash, Ph.D.

Academic Editor

PLOS ONE

Reviewers' comments:

Reviewer's Responses to Questions

**Comments to the Author**

1. If the authors have adequately addressed your comments raised in a previous round of review and you feel that this manuscript is now acceptable for publication, you may indicate that here to bypass the “Comments to the Author” section, enter your conflict of interest statement in the “Confidential to Editor” section, and submit your "Accept" recommendation.

Reviewer #1: (No Response)

Reviewer #2: (No Response)

2. Is the manuscript technically sound, and do the data support the conclusions?

Reviewer #1: Yes

Reviewer #2: Partly

3. Has the statistical analysis been performed appropriately and rigorously? 

Reviewer #1: I Don't Know

Reviewer #2: Yes

4. Have the authors made all data underlying the findings in their manuscript fully available?

Reviewer #1: No

Reviewer #2: No

5. Is the manuscript presented in an intelligible fashion and written in standard English?

Reviewer #1: Yes

Reviewer #2: Yes

6. Review Comments to the Author

Reviewer #1: This paper documents a seemingly under-documented phenomena, that being maternal migration during pregnancy, and post-pregnancy, or as the authors name it “Temporary Childbirth Migration”. The authors use mixed methods to describe the phenomena of migrating during pregnancy or postpartum in two states in India.

Comments that myself and previous reviewer made in our last review were somewhat addressed, but there are still several comments which were not addressed sufficiently. My major concern is the continued (and increased) reference to analyses which was done and is never shown to the reader. I understand the authors preference for a compact paper, but these results could be presented in an appendix or supplementary material. I do not like the reference to analysis the reader is unable to access.

Below I have outlined comments which I consider to be major and more minor comments. I think these comments require addressing prior to publication.

MAJOR COMMENTS

-Section 2.1: The authors should include reasons why these states were chosen for this analysis, because it is not clear as to why these specific states were chosen above all others.

This point was not sufficiently addressed. State how the availability of comparable control districts was evaluated, was this done statistically? Also state what the reasoning was for the Ministry of Woman and Child Development and the funding agency to suggest these districts. The question of why these districts were chosen does pose some questions about credibility, and I think the authors need to be more explicit in their decision for choosing these states. The including of the comments they make are welcome, but are not sufficient.

- Section 2.1.1: Is there no other source of selective sampling apart from apart from higher probability of sampling non-migrating mothers for your approach? Do some mothers “slip through the net” of the AWC/AWW system? If some mothers do slip through the net, then who are these likely to be and how would that impact your results and findings? The entire sample comes from those that engage with the AWC/AWW system. I think this is a major source of sample selection, and is a point not sufficiently addressed or consider.

This point was not addressed at all. If, as the paper claims, the hypothesis is that those temporarily migrating are less likely to engage in the AWC/AWW system, then there is a major selection problem here, and this induces a major source of bias in your estimates. Given that a key contribution of the paper is to sample mothers who are usually missed in surveys, it is worth stating specifically how this paper addresses those limitations of previous work. How are these individuals recruited given that they usually fall through the net? What do you do to mitigate this?

- Lines 306-314: I made a similar comment in previous review, but it was not sufficiently addressed. This feels a little like you’re picking the goalposts that suit your results, either that or the language makes it confusing. The WHO recommend 8 ANC visits, of which how many should be in the last month? The Indian government recommend 4 visits, of which how many should be in the last month? Is the measure of 5+ ANC visits for the last month only, or entire pregnancy? The choice of dependent variable should at least align with one of these 4 guidelines. If your outcome variable is for the final month, then it should exactly align with WHO’s or the Indian governments recommendations for the final month, not for the entire pregnancy. If there is no guideline for the final month, but you want your dependent variable to be ANC visits in final month, then it should be a count variable, not a binary one. This analysis is confusing, and I’m not sure whether your outcome aligns with one of the guidelines. I don’t think a reasonable argument to make is that the Indian guideline for entire pregnancy is 4 visits, so we look to see if mothers had over 4 visits in the final month, because this is not what the guideline says from my understanding. It may be that I do not fully understand the guidelines, in which case the writing needs to be tidied up a little.

- I agree with Reviewer 2’s major comment 2 from previous review. Clustering does not address the issue that reviewer 2 raises and I find it a little difficult to believe that the results would be the same if fixed effects were included. I understand the authors do not want to include too many tables, but they may consider an appendix, because there are several analyses which are referred to throughout the text, but the reader has no access to them.

This comment is also applicable to Reviewer 2’s major comment 1 and 3. The constant reference to analysis that was done and a conclusion of “the results were the same” does, unfortunately, question the credibility of the analysis.

Indeed, this is a more general point that there are several instances throughout the paper that refer to additional analysis which the reader is not able to access. I would suggest this needs remedying and including an appendix would be a reasonable approach, given the authors preference for a more compact paper.

- Reviewer 2’s previous comment: “I am not very clear about propensity score matching mentioned in line 233 in section 2.1.1. Why was it used? What was the dependent variable?”

Although this was improved, I think the authors need to be marginally more explicit. Presumably the nearest neighbour matching was done, and some measure of error was used to choose the districts. But it’s still not clear why 852 villages were chosen, what was the criteria? Was it that 852 villages met your pre-set criteria, or was it that you wanted 852 villages and these ones were the best fitting ones?

MINOR COMMENTS

- I still find it a little peculiar that although there is at least anecdotal evidence of the ritual pollution, none of the survey respondents state that this was a reason for leaving. Although beyond the scope of this paper I think this is an important avenue for future research and exploration.

- Previous review comment: “Line 365: “Working outside of the home is associated with increased odds of returning to the natal home in Madhya Pradesh but reduces the odds of returning home in Bihar.”

My preference is for these differences to be discussed and rationalised, because at present it’s difficult to draw conclusions (esp. for policy makers) because the effect is exactly in the opposite direction for each state. I would prefer that these differences be explained to the reader, because it is perplexing; without some explanation the reader is unsure of what conclusions to draw. The state specific description doesn’t really help explain this result. I would prefer the authors make it easier for the reader to understand the strange result.

- It may be worth including a table similar to “Table 1: Demographics of sample” but instead by whether temporary migrated or not.

- Reviewer 2’s comment: “Tables 3, 4: Are the significance levels indicative of difference between states? Mention this detail in table footnotes.”

In would usually prefer that tables be readable on their own, without the need for inspecting the text. In other words, full details of how the numbers in the table were calculated should be included in the footnotes. Putting “*** p<0.01, ** p<0.05, * p<0.1” does not tell the reader what test was done to reach that result, was it the difference in means between states, or was it the test of coefficients being different from zero.

Reviewer #2: Temporary Childbirth Migration and Maternal Healthcare in India

The authors have revised their manuscript based on suggestions, but I don’t see the new results anywhere (although authors have added text related to the comments). At this stage I would like to request authors to provide all the results requested in a separate file or the main draft so that they can be assessed and properly included in the main draft.

Specifically, the following results should be provided and complemented with a detailed discussion of the results (i.e. how the coefficient of interest changes etc) and models which were used:

1) Replication of results in Table 2 with 3-month post-partum as the cut-off, 1-month post-partum as the cut-off and full-sample (where no cut-offs are used). Please provide explanation as to how the coefficient is stable despite changing the cut-offs which establishes the robustness of the result.

2) Show results where District fixed Effects have been used. Please add text which discusses how the estimates remain unchanged after the introduction of district fixed effects.

3) Provide results with selection model. Show results for both the first and second stage and provide a commentary on these results as well. Please provide exact empirical models/equations which were used to reduce any confusion, this will help in better understanding and will provide more clarity.

Suggestion: The selection model is a superior model of analysis in this set-up so rather than treating it as a robustness/sensitivity check, the main results can be replaced by the results from the selection model.

4) For results provided in Table 6: the main explanatory variable should be changed to “number of months spent in the natal home during pregnancy” in light of the explanation provided by the authors. This new variable will correctly capture the relationship between length of stay at natal home and care received in form of ANC visits. The older variable “being at natal home during last month of pregnancy” will not correctly capture the total number of ANC visits tied to the natal home.

For reference my previous comment and current authors response:

“5) Table 6: Explores the relationship “Five or more ANC visits” and “Being at natal home during last month of pregnancy”. Shouldn’t the dependent variable be number of ANC visits during last month of pregnancy. Most of the ANC visits in this scenario (where a mother only migrated during last month of the pregnancy) would have already taken place in her marital home rather than natal home.

>> Yes, we agree that some of these visits may have occurred before the woman migrated, however, what we know about care seeking in pregnancy in India (and elsewhere) is that few women have visits early in pregnancy, and the number of visits increases in later pregnancy. A different (and interesting) analysis could be to see if women who migrated early in pregnancy were more likely to have a visit in the first trimester (which is an important gap), however, we did not do so in this analysis because of sample size issues with the number of women migrating in early pregnancy and because we were interested in using a standard measure of appropriate/adequate ANC visits. We had the exposure be “being at the natal home in the last month of pregnancy” to make sure that we were comparing appropriate groups of women to each other. For example, if we looked at women who were at the natal home in month 6, they might not have had enough time to get the adequate number of visits. By looking at women in the last month of pregnancy we are able to ensure that all women had comparable time to have enough visits.”

7. PLOS authors have the option to publish the peer review history of their article (what does this mean?). If published, this will include your full peer review and any attached files.

Reviewer #1: No

Reviewer #2: **Yes: **Prachi Singh

---

## [Author Response · Author response to Decision Letter 1]

20 Mar 2022

March 19, 2022

To the editors and reviewers, 

Thank you again for the opportunity to revise and resubmit this paper, and for your very thoughtful and detailed comments. We have worked to again revise this paper, being more careful to include all new models and address the reviewer comments. 

Please find a point by point response below, and let us know if you have additional questions, 

Thank you again for your time and support,

Nadia Diamond-Smitha

Reviewer #1: This paper documents a seemingly under-documented phenomena, that being maternal migration during pregnancy, and post-pregnancy, or as the authors name it “Temporary Childbirth Migration”. The authors use mixed methods to describe the phenomena of migrating during pregnancy or postpartum in two states in India.

Comments that myself and previous reviewer made in our last review were somewhat addressed, but there are still several comments which were not addressed sufficiently. My major concern is the continued (and increased) reference to analyses which was done and is never shown to the reader. I understand the authors preference for a compact paper, but these results could be presented in an appendix or supplementary material. I do not like the reference to analysis the reader is unable to access.

Below I have outlined comments which I consider to be major and more minor comments. I think these comments require addressing prior to publication.

MAJOR COMMENTS

-Section 2.1: The authors should include reasons why these states were chosen for this analysis, because it is not clear as to why these specific states were chosen above all others.

This point was not sufficiently addressed. State how the availability of comparable control districts was evaluated, was this done statistically? Also state what the reasoning was for the Ministry of Woman and Child Development and the funding agency to suggest these districts. The question of why these districts were chosen does pose some questions about credibility, and I think the authors need to be more explicit in their decision for choosing these states. The including of the comments they make are welcome, but are not sufficient.

We have added more details to clarify and to increase the credibility of our research but adding the complete study design already covered in the protocol would make the paper lengthy. As such, we also have added the protocol paper as a supplementary material and request the readers to read that in detail. 

- Section 2.1.1: Is there no other source of selective sampling apart from apart from higher probability of sampling non-migrating mothers for your approach? Do some mothers “slip through the net” of the AWC/AWW system? If some mothers do slip through the net, then who are these likely to be and how would that impact your results and findings? The entire sample comes from those that engage with the AWC/AWW system. I think this is a major source of sample selection, and is a point not sufficiently addressed or consider.

This point was not addressed at all. If, as the paper claims, the hypothesis is that those temporarily migrating are less likely to engage in the AWC/AWW system, then there is a major selection problem here, and this induces a major source of bias in your estimates. Given that a key contribution of the paper is to sample mothers who are usually missed in surveys, it is worth stating specifically how this paper addresses those limitations of previous work. How are these individuals recruited given that they usually fall through the net? What do you do to mitigate this?

Yes, this is a limitation of this study. We have extended our discussion of this limitation in the paper as such “Our data also come from women who are registered at Anganwadi Centres, so the results are only generalizable to women from these districts who are registered in the ICDS system. The NFHS-4 in India, which was collected around the time of this study, found that 65% of women who gave birth in the last 5 years in Bihar and 81% in Madhya Pradesh were registered (5). Thus these findings can only be interpreted as describing maternal migration and associated impacts on care among a population of women who are already in the health care system during their pregnancies. Women who are never registered in pregnancy may be unlikely to receive care at all, and thus might face no impacts of migrating during pregnancy (since they were never seeking care). On the other hand, women who do not register in their husband’s home may migrate and somehow seek care in their natal home. More research is needed on this population.”

- Lines 306-314: I made a similar comment in previous review, but it was not sufficiently addressed. This feels a little like you’re picking the goalposts that suit your results, either that or the language makes it confusing. The WHO recommend 8 ANC visits, of which how many should be in the last month? The Indian government recommend 4 visits, of which how many should be in the last month? Is the measure of 5+ ANC visits for the last month only, or entire pregnancy? The choice of dependent variable should at least align with one of these 4 guidelines. If your outcome variable is for the final month, then it should exactly align with WHO’s or the Indian governments recommendations for the final month, not for the entire pregnancy. If there is no guideline for the final month, but you want your dependent variable to be ANC visits in final month, then it should be a count variable, not a binary one. This analysis is confusing, and I’m not sure whether your outcome aligns with one of the guidelines. I don’t think a reasonable argument to make is that the Indian guideline for entire pregnancy is 4 visits, so we look to see if mothers had over 4 visits in the final month, because this is not what the guideline says from my understanding. It may be that I do not fully understand the guidelines, in which case the writing needs to be tidied up a little.

Thank you for this comment. We agree that the guidelines are confusing and it is hard to know what the marker of appropriate ANC is given that the WHO and the Indian guidelines differ. We discussed these comments as a team, and decided to present the analysis as a continuous variable (number of ANC visits) so that we are not adding a value judgement to the appropriate number of visits. We have thus changed the table and the discussion in the text. We kept the figure in that shows number of ANC visits to give the reader more understanding of the situation with ANC visits in this population. 

- I agree with Reviewer 2’s major comment 2 from previous review. Clustering does not address the issue that reviewer 2 raises and I find it a little difficult to believe that the results would be the same if fixed effects were included. I understand the authors do not want to include too many tables, but they may consider an appendix, because there are several analyses which are referred to throughout the text, but the reader has no access to them.

This comment is also applicable to Reviewer 2’s major comment 1 and 3. The constant reference to analysis that was done and a conclusion of “the results were the same” does, unfortunately, question the credibility of the analysis.

Indeed, this is a more general point that there are several instances throughout the paper that refer to additional analysis which the reader is not able to access. I would suggest this needs remedying and including an appendix would be a reasonable approach, given the authors preference for a more compact paper.

We apologize for not adding these additional analyses earlier. We have added 3 new appendices which show the additional sensitivity analysis that were conducted. We have added district fixed effects in our models. 

- Reviewer 2’s previous comment: “I am not very clear about propensity score matching mentioned in line 233 in section 2.1.1. Why was it used? What was the dependent variable?”

Although this was improved, I think the authors need to be marginally more explicit. Presumably the nearest neighbour matching was done, and some measure of error was used to choose the districts. But it’s still not clear why 852 villages were chosen, what was the criteria? Was it that 852 villages met your pre-set criteria, or was it that you wanted 852 villages and these ones were the best fitting ones?

We have elaborated more on the sample size calculations and how the number of villages were calculated. More details on the propensity score matching approach are available in our protocol paper made available as supplementary material for the reader. 

MINOR COMMENTS

- I still find it a little peculiar that although there is at least anecdotal evidence of the ritual pollution, none of the survey respondents state that this was a reason for leaving. Although beyond the scope of this paper I think this is an important avenue for future research and exploration.

We agree that more research should be done on this. It is possible that some of the rationale behind the responses given in the quantitative survey could be interpreted as being because of ritual pollution (for example, “It is the cultural and social norm” or “Husband’s family asked me to go”).

- Previous review comment: “Line 365: “Working outside of the home is associated with increased odds of returning to the natal home in Madhya Pradesh but reduces the odds of returning home in Bihar.”

My preference is for these differences to be discussed and rationalised, because at present it’s difficult to draw conclusions (esp. for policy makers) because the effect is exactly in the opposite direction for each state. I would prefer that these differences be explained to the reader, because it is perplexing; without some explanation the reader is unsure of what conclusions to draw. The state specific description doesn’t really help explain this result. I would prefer the authors make it easier for the reader to understand the strange result.

There are many differences between the states which are somewhat complexing, we agree. We have removed a few of the sentences about state differences since that is confusing in places. While the results are the appropriate place for us to be interpreting these differences, in the discussion we do have a section talking about broad differences between the states, which we hope provides the reader with some information about why these differences exist. Future research could explore contextual differences in more depth. 

- It may be worth including a table similar to “Table 1: Demographics of sample” but instead by whether temporary migrated or not.

Thank you, we had felt that Table 2 explored the differences in the socio-demographics by different migration statuses, but we have included the table that you requested as an appendix and will let the editors decide if they would like both of these in the paper. 

- Reviewer 2’s comment: “Tables 3, 4: Are the significance levels indicative of difference between states? Mention this detail in table footnotes.”

In would usually prefer that tables be readable on their own, without the need for inspecting the text. In other words, full details of how the numbers in the table were calculated should be included in the footnotes. Putting “*** p<0.01, ** p<0.05, * p<0.1” does not tell the reader what test was done to reach that result, was it the difference in means between states, or was it the test of coefficients being different from zero.

Thank you, we have clarified that this table shows the results of logistic regression models by adding this to the title, along with “(Odds Ratios, 95% confidence intervals)”

Reviewer #2: Temporary Childbirth Migration and Maternal Healthcare in India

The authors have revised their manuscript based on suggestions, but I don’t see the new results anywhere (although authors have added text related to the comments). At this stage I would like to request authors to provide all the results requested in a separate file or the main draft so that they can be assessed and properly included in the main draft.

Specifically, the following results should be provided and complemented with a detailed discussion of the results (i.e. how the coefficient of interest changes etc) and models which were used:

1) Replication of results in Table 2 with 3-month post-partum as the cut-off, 1-month post-partum as the cut-off and full-sample (where no cut-offs are used). Please provide explanation as to how the coefficient is stable despite changing the cut-offs which establishes the robustness of the result.

We have added an appendix with the results of Table 2 for any migration run for the full model. We have clarified in our text that while there are small differences, they do not change the direction or interpretation of data (“no meaningful differences in the interpretation of the findings (Appendix 2).” We do not think that it makes sense to show the analyses for the other cut offs (3 months, 1 month) because many women are still at their natal home during this time, and thus not captured in our data, and thus just showing these results would be biased. Showing all the data combined shows that the smaller sample we selected to focus on for the rest of the analyses which we feel captures most of the return migrants is a reasonable choice, even though it restricts our sample size somewhat. 

2) Show results where District fixed Effects have been used. Please add text which discusses how the estimates remain unchanged after the introduction of district fixed effects.

We have added district fixed effects to table 2 data for any migration. As can be seen the addition of this to the model does not change the magnitude of the associations meaningfully or direction of any associations. We think that because we cluster at the village level that this might already account for given correlation between clusters and villages and districts. 

3) Provide results with selection model. Show results for both the first and second stage and provide a commentary on these results as well. Please provide exact empirical models/equations which were used to reduce any confusion, this will help in better understanding and will provide more clarity.

Suggestion: The selection model is a superior model of analysis in this set-up so rather than treating it as a robustness/sensitivity check, the main results can be replaced by the results from the selection model.

We have revised our primary analyses for the impact of temporary childbirth migration on our four health-related outcomes to selection models, including results for both the first and second stage. 

4) For results provided in Table 6: the main explanatory variable should be changed to “number of months spent in the natal home during pregnancy” in light of the explanation provided by the authors. This new variable will correctly capture the relationship between length of stay at natal home and care received in form of ANC visits. The older variable “being at natal home during last month of pregnancy” will not correctly capture the total number of ANC visits tied to the natal home.

Apologies if our last response was not clear and led to more confusion. Unfortunately we did not collect data on each month of pregnancy and where the woman was, instead, as can be seen in Figure 3, months 0-3 and 4-6 were clumped together, and thus we only know if they spent any of those months at their natal home and we could not create a total number of months variable. The WHO recommendations are that 3 of the 8 ANC visits are supposed to be in the last month of pregnancy, and 2 in the second to last month. Based on the reviewer’s feedback, we created a new variable that summed the number of months that a woman was out of the last trimester (months 7, 8 or 9) and ran new models, which we have now included in the paper. These results do not change our interpretation of any of the relationships, but may be a better measure of the exposure. Thank you for this comment. 

For reference my previous comment and current authors response:

“5) Table 6: Explores the relationship “Five or more ANC visits” and “Being at natal home during last month of pregnancy”. Shouldn’t the dependent variable be number of ANC visits during last month of pregnancy. Most of the ANC visits in this scenario (where a mother only migrated during last month of the pregnancy) would have already taken place in her marital home rather than natal home.

Yes, we agree that some of these visits may have occurred before the woman migrated, however, what we know about care seeking in pregnancy in India (and elsewhere) is that few women have visits early in pregnancy, and the number of visits increases in later pregnancy. A different (and interesting) analysis could be to see if women who migrated early in pregnancy were more likely to have a visit in the first trimester (which is an important gap), however, we did not do so in this analysis because of sample size issues with the number of women migrating in early pregnancy and because we were interested in using a standard measure of appropriate/adequate ANC visits. We had the exposure be “being at the natal home in the last month of pregnancy” to make sure that we were comparing appropriate groups of women to each other. For example, if we looked at women who were at the natal home in month 6, they might not have had enough time to get the adequate number of visits. By looking at women in the last month of pregnancy we are able to ensure that all women had comparable time to have enough visits.”

---

## [Decision Letter · Decision Letter 2]

9 May 2022

PONE-D-21-16583R2Temporary childbirth migration and maternal health care in IndiaPLOS ONE

Dear Dr. Smith,

Thank you for submitting your manuscript to PLOS ONE. I sent this paper back to same reviewers and got mixed opinion. While R1 recommended me to accept, R2 recommended a rejection. I read the paper and I felt more aligned to R2 (esp. concerns around cherry picking results). I strongly encourage you to go through R2’s detailed comments — where he/she highlighted what was addressed and what was not — and address everything carefully. In addition, R1 has additional suggestions which will improve how the paper reads. I thank R1 and R2 for making such thoughtful comments and I hope you appreciate the same. Once you address suggestions from R1 and R2, the paper will be much more convincing. Given the time spent in revising a paper, I want to give you another opportunity to fully address all the points raised by R2. Hope you can do this carefully. 

We look forward to receiving your revised manuscript.

Kind regards,

Nishith Prakash, Ph.D.

Academic Editor

PLOS ONE

Reviewers' comments:

Reviewer's Responses to Questions

**Comments to the Author**

1. If the authors have adequately addressed your comments raised in a previous round of review and you feel that this manuscript is now acceptable for publication, you may indicate that here to bypass the “Comments to the Author” section, enter your conflict of interest statement in the “Confidential to Editor” section, and submit your "Accept" recommendation.

Reviewer #1: (No Response)

Reviewer #2: (No Response)

2. Is the manuscript technically sound, and do the data support the conclusions?

Reviewer #1: Yes

Reviewer #2: Partly

3. Has the statistical analysis been performed appropriately and rigorously? 

Reviewer #1: Yes

Reviewer #2: No

4. Have the authors made all data underlying the findings in their manuscript fully available?

Reviewer #1: Yes

Reviewer #2: No

5. Is the manuscript presented in an intelligible fashion and written in standard English?

Reviewer #1: Yes

Reviewer #2: No

6. Review Comments to the Author

Reviewer #1: Overall, I think this paper is interesting and does paper does contribute to the literature by documenting an under investigated phenomena. The paper does contribute a documentation of an under investigated phenomena which potentially has some profound implications for policy. The analysis, although limited in scope does appear to be a reasonable approach. The discussion and conclusion do follow from the results. However, the paper does not extend much beyond documenting this phenomenon. Given that the main contribution of the paper is to document the phenomena, it may be worthwhile emphasising why this is important to document and what implications these results have, especially for health outcomes.

I can see that documenting this phenomenon is potentially important for policymakers and for future research. For researchers, not knowing that mothers migrate during pregnancy or soon thereafter, is likely causing survey sampling issues because they migrate. Also, the authors make clear that many policies designed to support mothers during pregnancy are not designed to accommodate maternal migration, and therefore is important to document.

Overall, the authors have addressed reviewers’ comments. I am happy with the changes that were made, and the authors have improved their responses, compared to last time. However, there are still a handful of comments that need addressing prior to publication. These are small details but are necessary for publication.

The authors state that three items were included in the appendix, however I see only 2 tables. Specifically, the item which was requested “Appendix 1: Sociodemographic Characteristics of women who migrated at all compared to women who did not migrate at any point” is not included and missing, although the caption is included. Authors need to include this table in the appendix.

In Appendix Table 3, it is unclear what is meant by “district-level variable” and I don’t understand what the coefficient on this variable means. It was district fixed effects which were requested; therefore, I would not expect “district” to be included in the table with a coefficient. The reviewers requested a district fixed effect regression, and therefore the reviewers need to estimate this model and include this in the appendix. Usually, a district-level fixed effect model would not include “district” as an additional variable with a coefficient in the table .

Table 6 is unreadable, as it spans outside of the range of the page, and therefore I couldn’t read it. Authors need to make sure that pages with large tables, such as these, are either landscape, or scaled so that they are readable.

This comment was not addressed:

“Reviewer 2’s comment: “Tables 3, 4: Are the significance levels indicative of difference between states? Mention this detail in table footnotes.”

In would usually prefer that tables be readable on their own, without the need for inspecting the text. In other words, full details of how the numbers in the table were calculated should be included in the footnotes. Putting “*** p<0.01, ** p<0.05, * p<0.1” does not tell the reader what test was done to reach that result, was it the difference in means between states, or was it the test of coefficients being different from zero.”

Changes were made to other tables, but not to Tables 3 and 4. Again, what do these stars denote? Is this the significance in a t-tests of differences in means? Whatever the test used, include in footnote exactly what test was done to get these p-values. Be precise about where these p-values come from. The authors need to include precise detail of what test of differences was done, and state that they are testing the difference between states, in the footnote of Tables 3 and 4.

Reviewer #2: (No Response)

7. PLOS authors have the option to publish the peer review history of their article (what does this mean?). If published, this will include your full peer review and any attached files.

Reviewer #1: No

Reviewer #2: No

---

## [Author Response · Author response to Decision Letter 2]

4 Sep 2022

September 1, 2022

To the reviewers and editors of PlosOne, 

Thank you once again for your very thoughtful and detailed feedback on this manuscript. We greatly appreciate the energy you have put into helping us make this a strong and methodologically sound paper. Below we address each comment point-by-point. 

Thank you for your time, 

Nadia Diamond-Smith

Temporary Childbirth Migration and Maternal Healthcare in India 

Authors have tried to address many of the issues which were raised in the previous rounds however many of these were not sufficiently addressed, the draft lacks clarity in multiple areas and requires significant amount of rework. Comments which have been raised before: Please note I follow the same numbering as the one in my previous review (Items 1 to 5) 

1) Different cut-off points for analysis (Table 2): Authors have addressed this partially, but they have not shown results for 1-month/3-month post-partum cut-off. They mentioned possible bias in the results (I am unclear about their reasoning/argument) but it is still desirable to see these results even if they are not a part of the final draft but only provided to the reviewer as part of author’s response. Looking at these results will convince us that the authors are not cherry picking which analysis to show. 

We apologize for any continued confusion and concern regarding our analysis. However, we think that the reviewer’s comment is due to confusion about our analytic sample rationale. We have revised our analysis so that we use a consistent analytic sample throughout the paper of women 6-12 months postpartum. We maintain our rationale for the selection of women 6 or more months postpartum in the methods section, describing that this is to ensure that our sample is representative of all postpartum women, including those who stayed longer at their natal home. If we were to define our sample using a 1 month or 3 month postpartum cutoff, our sample would be biased towards women who stay a shorter period, do not migrate, or some other factors. However, by focusing on women who are back at their marital home by 6 months postpartum, we have data on women who returned earlier, but we also have data on women who returned later, thereby reducing any bias towards women with short stays or who do not migrate at all. 

2) District Fixed Effects: This comment has been addressed in Table 6 and has been incorporated incorrectly in Table 2 (results show one single variable “Districts” rather than dummy variables for each district). 

We have revised our regression models to a mixed-effects modeling strategy incorporating the two hierarchical effects, village and district levels, to accommodate for clustering. A population-averaged model better fits our analytic interest in the role of district within the current analysis, which is to accommodate for our sampling structure for each state, and not to explicitly model district-specific differences in relationships. Our text and tables have been modified to the new analysis. 

3) Selection problem: [Table 2 & 6 results will be directly affected by this analysis]: This concern has been raised by both the reviewers. Even though I had explicitly requested the mathematical model (first stage – which models the binary decision to migrate and second stage which models the outcomes we are interested in) and description of models/coefficients, authors haven’t provided these in detail and in fact some of description of the models is insufficient. Below is authors description of first stage (lines 358-359, 367- 371): “….generalized structural equation models to accommodate for potential selection effects into each temporary childbirth migration variable by sociodemographic characteristics.” “First stage models in joint analyses for temporary childbirth migration outcomes employed Poisson regression for the number of months spent in the natal home during the last trimester of pregnancy and logistic regression for natal home at time of childbirth for delivery analyses and natal home during the first month postpartum, also following above empirical models.” - What was/were the variables used in the first stage which was/were not included in the second stage (refer to Heckman model)? - How come the first stage is poisson model in case of a selection model (as mentioned in lines 367-371)? Shouldn’t it be a logistic model in all cases? [Please note that my original comment#4 in review round#2 was suggested when you were NOT using a selection model.] - Where are the results for first stage regression (in one of the many panels of Table 6?)? Where have they been discussed in detail in the manuscript? - The language is also quite confusing. I would suggest rewording this content for clarity. 

We appreciated the reviewer’s recommendation to incorporate our two-stage analysis with a joint modeling approach and have operationalized this analysis within a generalized structural equation modeling (SEM) approach. This broader approach was selected as it allows for us to model our count and binary outcomes appropriately through the use of Poisson and logistic regression models, respectively. While the Heckman selection model is a special case of generalized SEM modeling, this modeling approach is limited in that it cannot accommodate non-binary outcomes and its application is limited to selection due to censoring/non-observation of data. We have also added further detail to the methods section and results section as requested by the reviewer. 

4) Outcome choice in Table 6: First of all, this table in its current form in unreadable. The table doesn’t fit the page, different panels have been thrown together without appropriate labelling, the text description is unclear. What is the sample size for this analysis? Outcomes chosen by authors for Table 6: Number of ANC visits – This was observed at which time-window, the last month of pregnancy (as you mentioned in your response)? Please mention this in brackets as well in the results section so that clarity is maintained. Place of delivery – This choice is fine as it doesn’t depend on any particular choice of time window. Number of post-partum visits – Was this asked in the first month postpartum (as mentioned in line 311?)? Isn’t that shorter/different from the Indian guideline which is meant for 42 days rather than a month (line 330)? 

The analytic sample for Table 6 analyses is limited to women who were 6 or more months postpartum at the time of interview, which is now the consistent sample for all data presented in the paper. Our outcomes are specified in the measures section of our manuscript. We have added the sample size to the top row of each column. 

5) Similar to item #4. # Another item which is still unaddressed (was raised by Reviewer#1 in the last round and previously by me as well) is describing WHY results are different for the two states. Authors describe the state of ICDS in these two states and mention few other statistics but as Reviewer#2 mentioned this apparent difference has still not been rationalized for the reader. 

We have edited this paragraph for further clarity on potential differences in the results between the two states.

The issues related to empirical analysis/writing/layout/lack of clarity remain in the manuscript. Particularly, the lack of clarity/convincing arguments is a recurring issue in the manuscript where the content doesn’t flow/connect across sections and thus creates confusion. 

Minor Comments: 

1) Reference # 5 & 52 are same, retain only one. 

Edited

2) Line 250 mentions “Village matched” this creates confusion as the content of the companion paper is not a part of this draft. Either describe this approach in a footnote or choose alternate wording. 

We have added a citation, so that the reader can get more information if desired. We felt it was important to provide some context on the larger study and design to help inform the interpretation and also be transparent about where this data comes from

3) Line 239-240: “We hypothesize that natal home migration will lead to a disruption in the continuum of care and therefore lower access and utilization of services (Figure 1).” Literature provided by the authors suggests a movement in either direction, how was this particular hypothesis (i.e. disruption) selected? 

Thank you for this point, we have added “An alternate hypothesis, in the opposite direction, is that women could receive better care at their natal home.”

4) Line 284: Replace “12m” with less than 12 months age 

Edited

5) Reference to Appendix items 1 & 3 missing in draft. 

-Apologies. Based on reviewer feedback we have now integrated appendix 1 into table 1, and removed appendix 3 because we changed our models. 

6) Table notes should be expanded, at this stage it is not possible to read the tables independent of the text. Describe the model used, dependent, key independent variable/s, mention what * represent i.e. what is the hypothesis being tested. 

We have revised each of our tables carefully in order to improve the clarity of our presentation, expanding our titles and adding notes to each table so that the tables are able to be understood independent of the text. 

7) Mention in text and table what is IQR and IRR (along with their full form). 

Added

8) The legend description of mathematical equation is incorrect. It should be βi and not β1-i.

Thank you for identifying this error; we have corrected the model specification. 

R2

Overall, I think this paper is interesting and does paper does contribute to the literature by documenting an under investigated phenomena. The paper does contribute a documentation of an under investigated phenomena which potentially has some profound implications for policy. The analysis, although limited in scope does appear to be a reasonable approach. The discussion and conclusion do follow from the results. However, the paper does not extend much beyond documenting this phenomenon. Given that the main contribution of the paper is to document the phenomena, it may be worthwhile emphasising why this is important to document and what implications these results have, especially for health outcomes. I can see that documenting this phenomenon is potentially important for policymakers and for future research. For researchers, not knowing that mothers migrate during pregnancy or soon thereafter, is likely causing survey sampling issues because they migrate. Also, the authors make clear that many policies designed to support mothers during pregnancy are not designed to accommodate maternal migration, and therefore is important to document. Overall, the authors have addressed reviewers’ comments. I am happy with the changes that were made, and the authors have improved their responses, compared to last time. However, there are still a handful of comments that need addressing prior to publication. These are small details but are necessary for publication. 

Thank you so much for your ongoing feedback.

The authors state that three items were included in the appendix, however I see only 2 tables. Specifically, the item which was requested “Appendix 1: Sociodemographic Characteristics of women who migrated at all compared to women who did not migrate at any point” is not included and missing, although the caption is included. Authors need to include this table in the appendix. 

-Apologies, we decided that this information was important and we moved it to Table 1 of the main manuscript.

In Appendix Table 3, it is unclear what is meant by “district-level variable” and I don’t understand what the coefficient on this variable means. It was district fixed effects which were requested; therefore, I would not expect “district” to be included in the table with a coefficient. The reviewers requested a district fixed effect regression, and therefore the reviewers need to estimate this model and include this in the appendix. Usually, a district- level fixed effect model would not include “district” as an additional variable with a coefficient in the table. 

Please see our response to reviewer 1, comment 2. Subsequently, we have removed this appendix. 

Table 6 is unreadable, as it spans outside of the range of the page, and therefore I couldn’t read it. Authors need to make sure that pages with large tables, such as these, are either landscape, or scaled so that they are readable. 

We have revised the layout of this table. 

This comment was not addressed: “Reviewer 2’s comment: “Tables 3, 4: Are the significance levels indicative of difference between states? Mention this detail in table footnotes.” In would usually prefer that tables be readable on their own, without the need for inspecting the text. In other words, full details of how the numbers in the table were calculated should be included in the footnotes. Putting *** p<0.” does not tell the reader what test was done to reach that result, was it the difference in means between states, or was it the test of coefficients being different from zero.”

Changes were made to other tables, but not to Tables 3 and 4. Again, what do these stars denote? Is this the significance in a t-tests of differences in means? Whatever the test used, include in footnote exactly what test was done to get these p-values. Be precise about where these p-values come from. The authors need to include precise detail of what test of differences was done, and state that they are testing the difference between states, in the footnote of Tables 3 and 4. 

-Apologies, we have added to our notes under each table that these represent the results of tests of differences in the distribution of these factors between the states using chi-squared tests. As noted in our responses to reviewer 1, we have also revised the titles and added further text below the tables to improve clarity.

---

## [Decision Letter · Decision Letter 3]

1 Dec 2022

PONE-D-21-16583R3Temporary childbirth migration and maternal health care in IndiaPLOS ONE

Dear Dr. Diamond-Smith,

Thank you for for the revised draft. I heard from the two referees and they all (so do I) appreciate the revised draft where you have addressed majority of the comments. However, R2 (and to some extent R1) have serious concerns around selection (see their comments). You need to address this and remaining concerns from R2 and R1 in the revised draft. I do plan to send this back to the referees, so I recommend you to take time to address these comments carefully and draft a detailed response. 

We look forward to receiving your revised manuscript.

Kind regards,

Nishith Prakash, Ph.D.

Academic Editor

PLOS ONE

Reviewers' comments:

Reviewer's Responses to Questions

**Comments to the Author**

1. If the authors have adequately addressed your comments raised in a previous round of review and you feel that this manuscript is now acceptable for publication, you may indicate that here to bypass the “Comments to the Author” section, enter your conflict of interest statement in the “Confidential to Editor” section, and submit your "Accept" recommendation.

Reviewer #1: (No Response)

Reviewer #2: (No Response)

2. Is the manuscript technically sound, and do the data support the conclusions?

Reviewer #1: Yes

Reviewer #2: Partly

3. Has the statistical analysis been performed appropriately and rigorously? 

Reviewer #1: Yes

Reviewer #2: No

4. Have the authors made all data underlying the findings in their manuscript fully available?

Reviewer #1: No

Reviewer #2: No

5. Is the manuscript presented in an intelligible fashion and written in standard English?

Reviewer #1: Yes

Reviewer #2: Yes

6. Review Comments to the Author

Reviewer #1: Major Comments

The selection model comments made by Reviewer 2 (comment 3 in round 3) has not been addressed. My interpretation of the back-and-forth between the review and author is that the authors do not understand the comments being made, either that, or the discussion in the paper text are not sufficiently clear to describe the analysis. The problem that Reviewer 2 is outlining is that the second stage includes only those individuals that migrated, which is a non-random sample of the population, and including only migrants induces bias in your analysis from this non-random selection. The purpose of the selection model is to reduce the bias caused by the analysis of a non-randomly selected sample. The procedure is as follows (for heckman selection):

1) Estimate the probability of being in the sample (using all observations including those that don’t appear in your secondary analysis) using a probit,

2) Include the inverse mills ratio -- which is calculated using the probabilities from the first stage -- in your second stage equation. I believe your second stage equation are your main results, however I’m not sure if they are sample selection corrected by including the inverse mills ratio or equivalent.

There are alternatives to the heckman selection, however sample selection models are usually two stage and the details of each stage should be made explicitly clear in the text. It is also usually the case that authors present estimates from the first stage in their analysis. Review 2 has asked that a sample selection model be carried out and that the first stage results are presented for the reader to see.

In response to authors’ comment: You can combine both a Heckman selection model with a Poisson regression in the second stage with the ‘heckpoisson’ command in Stata.

It is possible that we have misunderstood your model – which is why I usually prefer mathematical equations to be used when discussing quantitative analysis, as it makes it easier for the reader to understand the authors approach – however this suggests that the discussion of the analysis requires improvement, so that the reader can better understand what is being done. If a sample selection model is not required, because the sample is randomly selected, then this should be better stated in the text.

Minor Comments

Table 1 goes off the page. Reformat for publication.

Stars representing results of statistical tests are a little odd. In table 6 there appears to have been two tests conducted, however I’d usually prefer different tests to be presented in different tables. My suggestion is to have two separate tables those being: Differences between states and the corresponding test (i.e. column for each state), and separately, between migration status and corresponding test (i.e. column for each migration status). The current presentation is a little peculiar.

Looks like the fixed-effect analysis is consistent with the main results, I suggest mentioning that they are consistent in the main text.

Reviewer #2: Authors have indeed addressed many of the comments from the last round. The presentation of tables/tables has also signitficantly improved, making it easier to read them without the main text.

I will highlight areas/items which still need some work, some of the comments carry forward from the previous round (as they have been partially addressed) but I must highlight that all of the MINOR comments and MANY of the MAJOR comments have been handled in the current version.

Comments:

1) ESSENTIAL/MAJOR

a) Selection problem: The mixed effects generalized strutural model is used to address this in the current version of the manuscript. This approach uses a joint analysis for temporary child migration (a count or a binary variable) and health outcome (number of ANC visits, private delivery, postpartum care etc). The sociodemographic characteristics being used for modelling both the variables are identical. I still fail to understand how in absense of an exogenous variable this model is able to address the selection problem. The manuscript is also silent about this, and again this is something that I have raised before. The name of a model used doesn't automatically fill-in the gap about understanding HOW that model helps in addressing an issue (selection in this case).

b) District Fixed Effects: This is again a comment which has been raised before but has been left unaddressed. Authors instead use a mixed effects model (incorporating for hierarchical structure of villages embedded in districts). However clustering is not a perfect substitute for fixed effects. The idea is not to view (as authors responded in their letter) "district-specific differences in relationships" but rather to explicitly account for time invariant district specific characteristics and to see whether that changes the main relationships being studied in this paper. Having no transparency when it comes to these set of results does make me uncomfortable, since it should be pretty straight forward thing to implement.

c) The current write up of Section 2.1.3 Analysis needs some more work. Equations for different outcomes should use different notations rather than identifical ones which creates confusion. Also not all subscripts have been defined, what is "j" in u0J (line 362)? What is "j" again in equation 2? what is Xij (line 365)? Beta1 as mentioned in line 365 is not temporary childbirth migration (TCM) dose but rather the impact (coefficient) of TCM on health outcomes. A line describing why we need cluster random effects will also be helpful.

2) NEED MORE DISCUSSION / CHANGES

a) Why do results not hold for postpartum care? Table 7 presents these results and discussion section mentions this without any supporting discussion.

b) Table 6 - how do you read this table? There are stars in all columns for few rows.

3) MINOR

a) Table 2 - how can the total on top of the column be calculated from rows below? Can you describe how the total for Madhya Pradesh (1296) be derived from rows below?

b) Line 329-330 menion "As can be seen in Figure 2 most women in India do not meet the Indian guidelines, and only 25% receive more than 4 ANC visits." Figure 2 is about percentage of women indulging in TCM and not about ANC visits.

c) Table 1 doesn't fit the page. Similar to other wide tables use widescape instead of portrait for this table

d) Couple of formatting errors (please carefully go through text again to be sure about any remaining errors like these):

i) Line 101 insert space before (9,10).

ii) Line 245 change to "...quantify the impact of Temporary Childbirth Migration on health service use"

iii) Line 253, change to "(52)."

iv) Line 255 insert space before bracket

v) Line 275-277: I can't understand the text currently. Perhaps remove or edit to improve readability.

vi) All places where beta1-k is mentioned should have the k in subscript rather than normal text.

vii) Line 573: correct to "...play out"

7. PLOS authors have the option to publish the peer review history of their article (what does this mean?). If published, this will include your full peer review and any attached files.

Reviewer #1: No

Reviewer #2: No

---

## [Author Response · Author response to Decision Letter 3]

1 Jun 2023

May 31, 2023

To the editor and reviewers, 

Thank you for the opportunity to revise and resubmit this paper, and apologies for the very delayed response time. We have very carefully detailed our response to the reviewers below. We would like to note that we involved a PhD-level biostatistician to assist us with reviewing and revising analysis and its description, due to the back and forth that we have had over the last few rounds of reviews. We carefully considered the feedback and have responded below to the very best of our ability. 

Reviewer comments in bold, our responses in regular text. Please let us know if you have additional feedback. 

Thank you again for your time and feedback, 

Nadia Diamond-Smith, on behalf of my co-authors

Reviewer #1: Major Comments

The selection model comments made by Reviewer 2 (comment 3 in round 3) has not been addressed. My interpretation of the back-and-forth between the review and author is that the authors do not understand the comments being made, either that, or the discussion in the paper text are not sufficiently clear to describe the analysis. The problem that Reviewer 2 is outlining is that the second stage includes only those individuals that migrated, which is a non-random sample of the population, and including only migrants induces bias in your analysis from this non-random selection. The purpose of the selection model is to reduce the bias caused by the analysis of a non-randomly selected sample. The procedure is as follows (for heckman selection):

1) Estimate the probability of being in the sample (using all observations including those that don’t appear in your secondary analysis) using a probit,

2) Include the inverse mills ratio -- which is calculated using the probabilities from the first stage -- in your second stage equation. I believe your second stage equation are your main results, however I’m not sure if they are sample selection corrected by including the inverse mills ratio or equivalent.

There are alternatives to the heckman selection, however sample selection models are usually two stage and the details of each stage should be made explicitly clear in the text. It is also usually the case that authors present estimates from the first stage in their analysis. Review 2 has asked that a sample selection model be carried out and that the first stage results are presented for the reader to see.

In response to authors’ comment: You can combine both a Heckman selection model with a Poisson regression in the second stage with the ‘heckpoisson’ command in Stata.

It is possible that we have misunderstood your model – which is why I usually prefer mathematical equations to be used when discussing quantitative analysis, as it makes it easier for the reader to understand the authors approach – however this suggests that the discussion of the analysis requires improvement, so that the reader can better understand what is being done. If a sample selection model is not required, because the sample is randomly selected, then this should be better stated in the text.

We appreciate both reviewers’ guidance that further details are needed to clarify our modeling approach and our rationale for this analysis. In presenting our response, we would first like to share the back and forth between the authors and reviewers across the four rounds of review that this paper has undergone for context. We then describe the edits which have been made since our prior version. 

• In our first round of reviewer comments (September 2021), Reviewer 2, Comment 3 suggested a selection model as follows: 

Reviewer feedback: 3) How many women (in each state) out of the total sampled women participated in temporary childbirth migration? The analysis can be complemented with a selection model wherein in the first stage decision to migrate is modelled and in the second stage healthcare outcomes are modelled.

Author’s response: A total of 2,921 women overall participated in any temporary childbirth migration, 1,753 in Bihar and 1,168 in Madhya Pradesh. We have added these numbers to the text. 

We appreciate the reviewer’s suggestion for extending our analysis and have developed joint models incorporating selection into temporary childbirth selection for each of our outcomes of interest using a generalized structural equation modeling approach. We ran these models and there were no changes in the relationships between temporary childbirth migration and the outcomes, and thus we have not changed our models in the paper. We have however added that we ran this as a sensitivity analysis in the paper itself “We also ran joint models incorporating selection into temporary childbirth selection for each of our outcomes of interest using a generalized structural equation modeling approach as another sensitivity analysis. These models did not differ substantially from the models presented and thus are not shown.”

• In our second round of reviewer comments (March 2022), Reviewer 2 Comment 3 relates to the selection model, as follows: 

Reviewer feedback: 3) Provide results with selection model. Show results for both the first and second stage and provide a commentary on these results as well. Please provide exact empirical models/equations which were used to reduce any confusion, this will help in better understanding and will provide more clarity.

Suggestion: The selection model is a superior model of analysis in this set-up so rather than treating it as a robustness/sensitivity check, the main results can be replaced by the results from the selection model.

Author’s response: We have revised our primary analyses for the impact of temporary childbirth migration on our four health-related outcomes to selection models, including results for both the first and second stage.

• In our third round of reviewer comments (September 2022), Reviewed 1 Comment 3 relates to the selection model, as follows: 

Reviewer feedback: 3) Selection problem: [Table 2 & 6 results will be directly affected by this analysis]: This concern has been raised by both the reviewers. Even though I had explicitly requested the mathematical model (first stage – which models the binary decision to migrate and second stage which models the outcomes we are interested in) and description of models/coefficients, authors haven’t provided these in detail and in fact some of description of the models is insufficient. Below is authors description of first stage (lines 358-359, 367- 371): “….generalized structural equation models to accommodate for potential selection effects into each temporary childbirth migration variable by sociodemographic characteristics.” “First stage models in joint analyses for temporary childbirth migration outcomes employed Poisson regression for the number of months spent in the natal home during the last trimester of pregnancy and logistic regression for natal home at time of childbirth for delivery analyses and natal home during the first month postpartum, also following above empirical models.” - What was/were the variables used in the first stage which was/were not included in the second stage (refer to Heckman model)? - How come the first stage is poisson model in case of a selection model (as mentioned in lines 367-371)? Shouldn’t it be a logistic model in all cases? [Please note that my original comment#4 in review round#2 was suggested when you were NOT using a selection model.] - Where are the results for first stage regression (in one of the many panels of Table 6?)? Where have they been discussed in detail in the manuscript? - The language is also quite confusing. I would suggest rewording this content for clarity. 

Author’s response: We appreciated the reviewer’s recommendation to incorporate our two-stage analysis with a joint modeling approach and have operationalized this analysis within a generalized structural equation modeling (SEM) approach. This broader approach was selected as it allows for us to model our count and binary outcomes appropriately through the use of Poisson and logistic regression models, respectively. While the Heckman selection model is a special case of generalized SEM modeling, this modeling approach is limited in that it cannot accommodate non-binary outcomes and its application is limited to selection due to censoring/non-observation of data. We have also added further detail to the methods section and results section as requested by the reviewer. 

In response to the reviewers’ feedback, we have added additional explanatory text to the paper to clarify that our analysis was not designed to accommodate for selection in the sense of informative missing data, which is a traditional use of the Heckman model.1 Instead, we used it to accommodate for bias that may be due to unobserved factors that jointly influence both migration and the health outcomes. In acknowledgement of this complicated relationship and to reduce the likelihood of bias within our model of temporary childbirth migration and outcomes, we employ a joint modeling strategy to simultaneously answer the following research questions: 

• First: What is the influence of sociodemographic characteristics (predictors) on temporary childbirth migration (outcome)?

• Second: What is the influence of temporary childbirth migration (primary predictor) and sociodemographic factors (secondary predictors) on outcomes: number of antenatal care visits, private (vs. other) facility delivery, and postpartum health check (vs. none), after controlling for the direct relationship between sociodemographic factors and temporary childbirth migration. 

We implement this analytic approach within Stata’s generalized structural equation modeling (GSEM) procedure which accommodates our unique combination of endogenous covariates, hierarchical data structure, and outcome structures (i.e., both binary and count data).2 In re-examining our analysis based on the reviewer comments we identified an error in our statistical analysis code which excluded the individual level random effect linking the joint models that accommodate selection bias. This adjustment necessitated a revision to our modeling approach due to some challenges fitting these complex models with multiple levels of random effects and correlated error terms. To simplify our models, we moved district from a random effect to a fixed effect while maintaining village as a random effect, and we adjusted the approach used for covariate inclusion from an a priori strategy to based on minimum threshold of association of p<0.1 in unadjusted models. Finally, we monitored for evidence of selection within these models, and for models where selection was estimated to be 0, we exclude the individual selection parameter from the final model presented. This occurred for two models: 1) outcome of any facility delivery for Bihar and 2) private facility delivery for MP and is noted in the methods sections as well as under the table. 

Compared to our prior analyses, the revisions presented within this resubmission were quite similar. Our interpretation of the findings on the impact of temporary childbirth migration (number of months at natal home in 3rd trimester) on the number of antenatal care visits achieved for both Bihar and Madhya Pradesh did not change from our prior analysis (Table 7). Similarly, our analyses on the odds of postpartum health check by being at the natal home postpartum were qualitatively similar in our revised analyses (Table 7). There was one exception. Our analysis of the impact of temporary childbirth migration (at natal home during childbirth) on private facility delivery was similar to our prior analysis for Madhya Pradesh but different for Bihar (Table 7). Our prior analysis identified a 55% increased odds of private facility delivery with being at the natal home during childbirth which was borderline statistically significant, our revised analysis reflected a non-statistically significant decrease with a large confidence interval. 

Minor Comments

Table 1 goes off the page. Reformat for publication.

Edited

Stars representing results of statistical tests are a little odd. In table 6 there appears to have been two tests conducted, however I’d usually prefer different tests to be presented in different tables. My suggestion is to have two separate tables those being: Differences between states and the corresponding test (i.e. column for each state), and separately, between migration status and corresponding test (i.e. column for each migration status). The current presentation is a little peculiar.

Thank you for this feedback, we have split this into 2 tables as you suggested and agree that this is much clearer. They are now labeled Table 6a and Table 6b. 

Looks like the fixed-effect analysis is consistent with the main results, I suggest mentioning that they are consistent in the main text.

Please see our response to the reviewer’s earlier comment regarding selection bias where we note that we have revised our modeling strategy to include district fixed effects instead of random, given model complexity. 

Reviewer #2: 

Authors have indeed addressed many of the comments from the last round. The presentation of tables/tables has also significantly improved, making it easier to read them without the main text. I will highlight areas/items which still need some work, some of the comments carry forward from the previous round (as they have been partially addressed) but I must highlight that all of the MINOR comments and MANY of the MAJOR comments have been handled in the current version.

Thank you so much for this feedback. We continue to be extremely grateful to our reviewers for their dedication to improving our paper through ongoing thoughtful and detail-oriented review. We hope that in this fourth revision, we have satisfactorily resolved all major reviewers’ concerns. 

Comments:

1) ESSENTIAL/MAJOR

a) Selection problem: The mixed effects generalized structural model is used to address this in the current version of the manuscript. This approach uses a joint analysis for temporary child migration (a count or a binary variable) and health outcome (number of ANC visits, private delivery, postpartum care etc). The sociodemographic characteristics being used for modelling both the variables are identical. I still fail to understand how in absense of an exogenous variable this model is able to address the selection problem. The manuscript is also silent about this, and again this is something that I have raised before. The name of a model used doesn't automatically fill-in the gap about understanding HOW that model helps in addressing an issue (selection in this case).

Our revised models accommodate selection through explicitly incorporating correlated errors terms across the two models. Please see other information regarding the revisions made to our models in our response to reviewer 1’s first comment. 

b) District Fixed Effects: This is again a comment which has been raised before but has been left unaddressed. Authors instead use a mixed effects model (incorporating for hierarchical structure of villages embedded in districts). However, clustering is not a perfect substitute for fixed effects. The idea is not to view (as authors responded in their letter) "district-specific differences in relationships" but rather to explicitly account for time invariant district specific characteristics and to see whether that changes the main relationships being studied in this paper. Having no transparency when it comes to these set of results does make me uncomfortable, since it should be pretty straight forward thing to implement.

With recent modeling adaptations for our selection models incorporating district fixed effects (table 7), we have adapted table 3 to include district fixed effects for consistency. We have added some language to the results and discussion section noting some district-specific differences in migration behavior and outcomes and have added language to the discussion section referring to this as an area of interest for future research, as follows: “Indeed, our fixed-effects analyses identified some differences in migration behaviors and outcomes within district-specific comparisons which suggests that future research should further explore the role of district-specific contextual variables.” (p29)

c) The current write up of Section 2.1.3 Analysis needs some more work. Equations for different outcomes should use different notations rather than identical ones which creates confusion. Also not all subscripts have been defined, what is "j" in u0J (line 362)? What is "j" again in equation 2? what is Xij (line 365)? Beta1 as mentioned in line 365 is not temporary childbirth migration (TCM) dose but rather the impact (coefficient) of TCM on health outcomes. A line describing why we need cluster random effects will also be helpful.

Thanks for your feedback. We have expanded our explanations in this section to ensure that each parameter is clearly explained and we note the multi-level nature of our data to clarify the use of random effects to accommodate clustering. 

2) NEED MORE DISCUSSION / CHANGES

a) Why do results not hold for postpartum care? Table 7 presents these results and discussion section mentions this without any supporting discussion.

Please see the paragraph starting on line 608 “Our findings that postnatal care (PNC) is not impacted by location similarly contradict the perspectives expressed by the AWWs, who expressed difficulties in identifying and providing care to women temporarily in their natal homes. It is possible that cultural practices of women staying in their homes for extended periods of time (mostly up to 40 days) in the postnatal period contributes to this, making it harder for AWWs to access them or even know of their existence, regardless of whether they migrated or not. Also, PNC received by mothers within 48 hours after delivery is already low (57% and 42% in Madhya Pradesh and Bihar respectively) and additional PNC is infrequent in general, perhaps because it is not highly prioritized by the government system or families, and therefore any difference might be hard to detect 3. Prospective designs and more nuanced quantitative and qualitative data are needed to disentangle these findings.”

b) Table 6 - how do you read this table? There are stars in all columns for few rows.

Thank you for this feedback, we have split this into 2 tables as you suggested and agree that this is much clearer. They are now labeled Table 6a and Table 6b.

3) MINOR

a) Table 2 - how can the total on top of the column be calculated from rows below? Can you describe how the total for Madhya Pradesh (1296) be derived from rows below?

The top column is the total number of women in each state, so adding the numbers below will not sum to that number. However, it is the denominator used to calculate the percentages. For example, in the first row, 782 women migrated at all, which is 42.3% of the full sample (782/1849=42.3%) 

b) Line 329-330 menion "As can be seen in Figure 2 most women in India do not meet the Indian guidelines, and only 25% receive more than 4 ANC visits." Figure 2 is about percentage of women indulging in TCM and not about ANC visits.

Apologies, we had removed the wrong figure in a previous revision round, we have replaced this. 

c) Table 1 doesn't fit the page. Similar to other wide tables use widescape instead of portrait for this table

Edited

d) Couple of formatting errors (please carefully go through text again to be sure about any remaining errors like these):

Thank you, we have carefully read through our text again and have made some additional edits. 

i) Line 101 insert space before (9,10).

Added

ii) Line 245 change to "...quantify the impact of Temporary Childbirth Migration on health service use"

Edited

iii) Line 253, change to "(52)."

Edited

iv) Line 255 insert space before bracket

Edited

v) Line 275-277: I can't understand the text currently. Perhaps remove or edit to improve readability.

We have edited this section to read as below, we have also removed some of the superfluous information which was not needed for a reader to understand this sample for this specific analysis. 

“1200 respondents from 200 villages in each arm from each state were needed to detect a difference of 5-9 percentage points in the child health outcomes of interest in the parent study (discussed above) from the counterfactual levels between 10-50 percent with an intra-cluster correlation coefficient between 0.15—0.30 4”

vi) All places where beta1-k is mentioned should have the k in subscript rather than normal text.

We thank the reviewer for their identification of this typographical error and have corrected this across the methods section. 

vii) Line 573: correct to "...play out"

Edited

---

## [Decision Letter · Decision Letter 4]

26 Jul 2023

PONE-D-21-16583R4Temporary childbirth migration and maternal health care in IndiaPLOS ONE

Dear Dr. Diamond-Smith,

Thank you for submitting your manuscript to PLOS ONE. This paper has taken unusually long but thats also because the revisions have not been up to mark. The concern has been the same -- section 2.1.3 that talks about selection. The paper has improved, but it is not there yet. I really hope you can address it, else I will have no choice but to reject the paper. Therefore, we invite you to submit a revised version of the manuscript that addresses the points raised during the review process.

We look forward to receiving your revised manuscript.

Kind regards,

Nishith Prakash, Ph.D.

Academic Editor

PLOS ONE

Journal Requirements:

Reviewers' comments:

Reviewer's Responses to Questions

**Comments to the Author**

1. If the authors have adequately addressed your comments raised in a previous round of review and you feel that this manuscript is now acceptable for publication, you may indicate that here to bypass the “Comments to the Author” section, enter your conflict of interest statement in the “Confidential to Editor” section, and submit your "Accept" recommendation.

Reviewer #1: (No Response)

2. Is the manuscript technically sound, and do the data support the conclusions?

Reviewer #1: Yes

3. Has the statistical analysis been performed appropriately and rigorously? 

Reviewer #1: Yes

4. Have the authors made all data underlying the findings in their manuscript fully available?

Reviewer #1: Yes

5. Is the manuscript presented in an intelligible fashion and written in standard English?

Reviewer #1: Yes

6. Review Comments to the Author

Reviewer #1: For the Authors

Overall, the authors have addressed our comments in this most recent version of the paper. I think this version is substantially improved.

However, I think there are still remaining question marks over the selection model that reviewer 2 has requested. Jointly estimating –and allowing for correlated errors—the two steps (migration first stage and outcomes of interest in the second stage) is welcome. But it is still not clear how the selection model has been implemented precisely. As per my comment below, the writing in this section needs to be improved before publication, as it is difficult to read and unclear. Further, the authors need to better outline how their approach handles the issues that the selection model claims to solve.

If this comment is addressed then I think the paper should be published.

Writing

Section 2.1.3: In general, the ‘Analysis’ section is quite difficult to read and decode. I’m still finding it difficult to understand the precise details of how the models were estimated. Before publication I think the authors should take care to better lay-out their steps and how each table was estimated. One way in which this could be implemented is to have sub-sections for each outcome and then explain the models for each outcome. I think it would aid the readers understanding if descriptive results were separated from the regression results.

Further, I think the authors need to make it more clear how their approach deals with the selection issues that reviewer 2 has outlined. One approach may be to specifically state the issue with estimating the models without considering selection and then state how their approach handles this issue.

Minor Comments

Line 307: there is a close bracket unassociated with an open bracket.

Line 363: “endogeneity issues” is a bit vague, I’d like that to be elaborated on and made clearer what endogeneity issues are present.

7. PLOS authors have the option to publish the peer review history of their article (what does this mean?). If published, this will include your full peer review and any attached files.

Reviewer #1: No

---

## [Author Response · Author response to Decision Letter 4]

25 Aug 2023

August 24, 2023

Dear Dr. Prakash,

We greatly appreciate the editorial team and peer reviewers for their ongoing dedication to the improvement of our manuscript including their detailed review and thoughtful feedback. We have carefully examined each comment shared by the reviewer, consulted with our authorship team and PhD-level biostatistics advisor on response, and revised our manuscript accordingly. Please see our detailed notes below on the specific edits which have been made to the manuscript for each reviewer comment, with reviewer comments provided in bold and our responses in regular text. We have also made a few additional edits identified by our team during review, also in tracked changes. 

We hope that you are satisfied with our response to the outstanding reviewer concerns and look forward to your response and any further feedback. 

Sincerely,

Nadia Diamond-Smith, on behalf of all co-authors

 

Comments to the Author

1. If the authors have adequately addressed your comments raised in a previous round of review and you feel that this manuscript is now acceptable for publication, you may indicate that here to bypass the “Comments to the Author” section, enter your conflict of interest statement in the “Confidential to Editor” section, and submit your "Accept" recommendation.

Reviewer #1: (No Response)

2. Is the manuscript technically sound, and do the data support the conclusions?

Reviewer #1: Yes

3. Has the statistical analysis been performed appropriately and rigorously? 

Reviewer #1: Yes

4. Have the authors made all data underlying the findings in their manuscript fully available?

Reviewer #1: Yes

5. Is the manuscript presented in an intelligible fashion and written in standard English?

Reviewer #1: Yes

6. Review Comments to the Author

Reviewer #1: For the Authors

Overall, the authors have addressed our comments in this most recent version of the paper. I think this version is substantially improved.

We greatly appreciate this comment. 

However, I think there are still remaining question marks over the selection model that reviewer 2 has requested. Jointly estimating –and allowing for correlated errors—the two steps (migration first stage and outcomes of interest in the second stage) is welcome. But it is still not clear how the selection model has been implemented precisely. As per my comment below, the writing in this section needs to be improved before publication, as it is difficult to read and unclear. Further, the authors need to better outline how their approach handles the issues that the selection model claims to solve. If this comment is addressed then I think the paper should be published.

Writing

Section 2.1.3: In general, the ‘Analysis’ section is quite difficult to read and decode. I’m still finding it difficult to understand the precise details of how the models were estimated. Before publication I think the authors should take care to better lay-out their steps and how each table was estimated. One way in which this could be implemented is to have sub-sections for each outcome and then explain the models for each outcome. I think it would aid the readers understanding if descriptive results were separated from the regression results.

Further, I think the authors need to make it more clear how their approach deals with the selection issues that reviewer 2 has outlined. One approach may be to specifically state the issue with estimating the models without considering selection and then state how their approach handles this issue.

We have re-reviewed our analysis description and added additional organization, text, and references to improve the clarity of this section. Our analysis section is now divided into five distinct sections using subheaders: 1) descriptive analyses of study participants and temporary childbirth migration practices, 2) regression modeling of sociodemographic characteristics and temporary childbirth migration, 3) descriptive analyses of reasons for temporary childbirth migration, 4) descriptive analyses of perinatal health care outcomes, and 5) regression modeling of temporary childbirth migration and perinatal health outcomes. Within each section we have expanded the text to clearly state what each analysis seeks to accomplish, including additional explanation on the potential concerns regarding endogeneity, and refer to each relevant results table to ensure that readers are able to link each analysis description with each result. We hope that these edits more clearly convey the purpose and method of each of the analyses undertaken in this manuscript. 

In addition, we have uploaded our data and analytic files to GitHub and reference them in the methods section of our manuscript (https://git.ucsf.edu/alison-elayadi/temporary-childbirth-migration.git) to ensure that readers of our manuscript have access to all supporting materials to understand our quantitative analyses. 

Minor Comments

Line 307: there is a close bracket unassociated with an open bracket.

Thank you for identifying this typographical error in our manuscript. The unassociated bracket has been removed. 

Line 363: “endogeneity issues” is a bit vague, I’d like that to be elaborated on and made clearer what endogeneity issues are present.

As noted above, we have expanded the text in this section to the following: “We selected this approach to accommodate the potential for endogeneity issues whereby certain unobserved factors (e.g., socioeconomic status, health status, etc.) might jointly influence both the likelihood of temporary childbirth migration and the perinatal health outcome, introducing bias into our estimation of this key relationship of interest. To reduce the risk of endogeneity bias, we allowed the error terms in the two equations to be correlated to monitor and mitigate selection bias in those that chose temporary migration.” p16 lines 387-392

7. PLOS authors have the option to publish the peer review history of their article (what does this mean?). If published, this will include your full peer review and any attached files.

Do you want your identity to be public for this peer review? For information about this choice, including consent withdrawal, please see our Privacy Policy.

Reviewer #1: No

---

## [Decision Letter · Decision Letter 5]

29 Sep 2023

Temporary childbirth migration and maternal health care in India

PONE-D-21-16583R5

Dear Dr. Diamond-Smith,

We’re pleased to inform you that your manuscript has been judged scientifically suitable for publication and will be formally accepted for publication once it meets all outstanding technical requirements.

Kind regards,

Nishith Prakash, Ph.D.

Academic Editor

PLOS ONE

Additional Editor Comments (optional):

Please fix the following issue: The GitHub link they include in the paper doesn’t work – is it possible that it only works within the UCSF network? It would be worth making sure this link works correctly prior to publication.

Reviewers' comments:

Reviewer's Responses to Questions

**Comments to the Author**

1. If the authors have adequately addressed your comments raised in a previous round of review and you feel that this manuscript is now acceptable for publication, you may indicate that here to bypass the “Comments to the Author” section, enter your conflict of interest statement in the “Confidential to Editor” section, and submit your "Accept" recommendation.

Reviewer #1: All comments have been addressed

2. Is the manuscript technically sound, and do the data support the conclusions?

Reviewer #1: Yes

3. Has the statistical analysis been performed appropriately and rigorously? 

Reviewer #1: Yes

4. Have the authors made all data underlying the findings in their manuscript fully available?

Reviewer #1: Yes

5. Is the manuscript presented in an intelligible fashion and written in standard English?

Reviewer #1: Yes

6. Review Comments to the Author

Reviewer #1: I think the authors have made a substantial effort in addressing previous comments, both in my previous review, but also comments from older reviews of this manuscript. I think this version has improved substantially from original version and the authors have been much more transparent. The authors have responded to my previous review by making the necessary changes to the paper. All my previous comments have been addressed. I believe that the manuscript now meets the necessary criteria and standard for acceptance. I have recommended for this paper to be accepted and I believe that the authors have made a substantial effort in addressing previous comments.

The only, small issue is that the GitHub link they include in the paper doesn’t work – is it possible that it only works within the UCSF network? It would be worth making sure this link works correctly prior to publication.

7. PLOS authors have the option to publish the peer review history of their article (what does this mean?). If published, this will include your full peer review and any attached files.

Reviewer #1: No

---

## [Editor Report · Acceptance letter]

31 Oct 2023

PONE-D-21-16583R5 

Temporary childbirth migration and maternal health care in India 

Dear Dr. Diamond-Smith:

I'm pleased to inform you that your manuscript has been deemed suitable for publication in PLOS ONE. Congratulations! Your manuscript is now with our production department. 

Kind regards, 

on behalf of

Professor Nishith Prakash 

Academic Editor

PLOS ONE